# Universal Reasoner: A Single, Composable Plug-and-Play Reasoner for Frozen LLMs

## Abstract

Large Language Models (LLMs) have demonstrated remarkable general capabilities, but enhancing skills such as reasoning often demands substantial computational resources and may compromise their generalization. While Parameter-Efficient Fine-Tuning (PEFT) methods offer a more resource-conscious alternative, they typically require retraining for each LLM backbone due to architectural dependencies. To address these challenges, here we propose Universal Reasoner (UniR) - a single, lightweight, composable, and plug-and-play reasoning module that can be used with larger frozen LLMs to endow it with specialized reasoning capabilities. Specifically, UniR decomposes the reward into a standalone reasoning module that is trained in a decoupled manner using predefined rewards, effectively translating trajectory-level signals into token-level guidance. Once trained, UniR can be combined with frozen LLMs at inference time by simply adding its output logits to those of the LLM backbones. This additive structure naturally enables modular composition: multiple UniR modules trained for different tasks can be jointly applied by summing their logits, enabling complex reasoning via composition. Experiments on mathematical reasoning and machine translation show that UniR surpasses existing baseline fine-tuning methods. Furthermore, UniR demonstrates weak-to-strong generalization: reasoning modules trained on smaller models effectively guide much larger LLMs. Beyond this, UniR generalizes across modalities such as in vision language models and domains such as medical reasoning. This makes UniR a cost-efficient, adaptable, and robust solution for enhancing reasoning in LLMs without compromising their core capabilities.

## 1 Introduction

Recent Large Language Models (LLMs), such as OpenAI-o1 (OpenAI, 2024), DeepSeek-R1 (DeepSeek-AI, 2025), Kimi-1.5 (Team et al., 2025), BOLT (Pang et al., 2025), and Open-Reasoner-Zero (Hu et al., 2025), have demonstrated remarkable capabilities in complex reasoning tasks. These models leverage Chain-of-Thought (CoT) (Wei et al., 2022) to elicit step-by-step reasoning processes from LLMs, thereby significantly enhancing their ability to tackle intricate problems. A key driver of this progress has been Reinforcement Fine-Tuning (RFT), which aligns model outputs with desired criteria, often human preferences, through reinforcement learning algorithms such as PPO (Schulman et al., 2017), DPO (Rafailov et al., 2023), GRPO (Shao et al., 2024), etc.

Unfortunately, adapting these powerful LLMs to specialize in specific domains or further enhance their reasoning abilities through RFT presents significant hurdles. The process is often prohibitively expensive due to the substantial computational and memory resources required for fine-tuning large-scale models. While Parameter-Efficient Fine-Tuning (PEFT) techniques like LoRA (Hu et al., 2022) have been proposed to mitigate these costs, they introduce their own limitations: (1) Model Dependency: LoRA modules strongly depend on specific model architectures, limiting their transferability across models of different sizes (e.g., from 3B to 14B). (2) Poor Linearity Support: Combining multiple domain-specific LoRA adapters through linear operations lacks robust theoretical support and frequently yields unpredictable outcomes (Wang et al., 2023).

To address these challenges, we introduce **UniR** (**Uni**versal **R**easoner), a novel method that decouples the reasoning capability into a specialized reasoning module, independently trained and

designed to enhance the capabilities of a backbone LLM. This lightweight reasoning module can be seamlessly integrated into a backbone model with a compatible tokenizer, without requiring explicit information about the model's internal structure. This approach not only fosters modularity, preserving the core capabilities of the base model, but also enables effective weak-to-strong generalization where a compact reasoning module can significantly enhance larger models. Moreover, this modularity opens avenues for combining distinct reasoning modules across multiple domains such as visual mathematical reasoning and medical reasoning.

The core innovation of UniR lies in its decoupling of reward model training from policy updates. While many alignment methods rely on preference datasets, UniR is designed to leverage explicit, predefined rewards that are often available for tasks with verifiable solutions, such as mathematical problem-solving or machine translation. We propose to model these typically trajectory-level rewards, or rewards over a full response, as a sum of log-probabilities derived from our decomposed reasoning module. This formulation effectively translates a global reward signal into a sequence of implicit token-level guidance signals that steer the reasoning process. The reasoning module is then trained to maximize this reward using a policy gradient algorithm, ensuring that only the smaller reasoning module is updated while the backbone policy remains unchanged. During inference, reasoning module provides guidance to a main backbone model by combining their outputs, typically by adding their respective logits before sampling. Furthermore, because the guidance occurs at the logit level, multiple reasoning modules trained for distinct tasks can be composed through simple logit addition—enabling composite probabilistic reasoning without additional training or architectural modification.

Our contributions are summarized as follows:

- **Computationally Efficient Training.** We introduce a method to train a decomposed reasoning module by optimizing verifiable rewards via token-level, policy-gradient guidance. Furthermore, our algorithm backpropagates only through a smaller reasoning model while keeping the large backbone model frozen, reducing computational and memory overhead.
- **Transferability Across Model Sizes.** Although our approach trains only small models, the learned reasoning module can be effectively transferred to larger models at inference time. For example, a 1.5B reasoning module trained alongside a 3B backbone can be used in conjunction with a 14B model, providing token-level guidance that improves the larger model's performance without requiring any fine-tuning of the 14B model itself.
- **Composability.** Our method allows seamless composition of multiple small reasoning modules specialized for specific domains or tasks. These specialized models can then be selectively deployed based on the domain context, enabling modular and efficient inference pipelines where domain expertise is encapsulated in lightweight, task-specific policies.
- **Superior Performance on Diverse Tasks.** Our experiments show that UniR substantially outperforms conventional fine-tuning methods, showcasing its efficacy in enhancing LLM capabilities on challenging mathematics, translation, and even medical benchmarks.

## 2 RELATED WORK

**Enhancing LLM Reasoning.** Improving LLM reasoning is key for complex reasoning and understanding (Durante et al., 2024). One prominent strategy involves Supervised Fine-Tuning (SFT) on specialized, often small, datasets curated to foster reasoning skills; these datasets might be distilled from larger models (Open Thoughts, 2025) or meticulously selected from high-quality examples (Muennighoff et al., 2025; Ye et al., 2025). Recently, Reinforcement Learning (RL)-based Group Relative Policy Optimization (GRPO) and related methods (Shao et al., 2024; Liu et al., 2025b) have shown promise in enhancing LLM reasoning capabilities using rule-based rewards without requiring a separate value function. Furthermore, the concept of verifiable rewards is extended by using generative models to provide soft reward signals for unstructured tasks (Su et al., 2025b). Our work leverages verifiable rewards to improve the reasoning ability of LLMs using the specialized reasoning module on tasks that require complex reasoning.

**Efficient Methods for Domain-Specific Tasks.** To mitigate the substantial computational costs of training LLMs, Parameter-Efficient Fine-Tuning (PEFT) methods like LoRA (Hu et al., 2022) are widely used, but their modules are tied to the model architecture, limiting transferability across

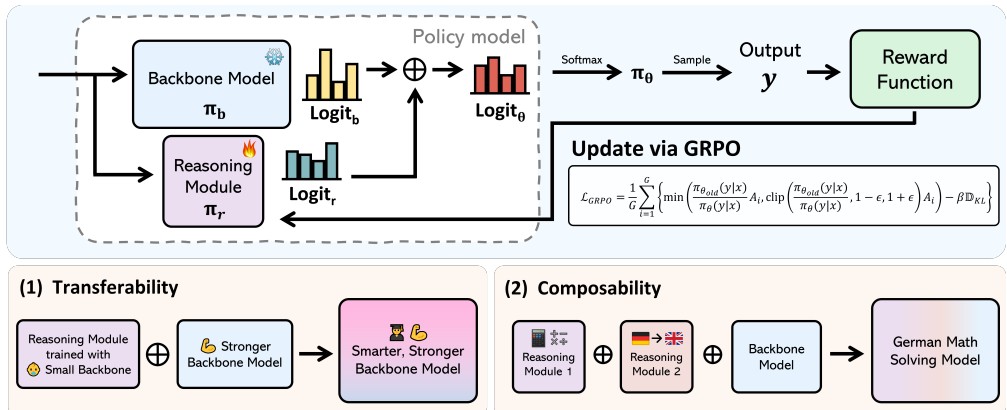

Figure 1: **UniR Framework Overview**. Our approach trains a lightweight, transferable reasoning module ($\pi_r$) using predefined rewards to guide a frozen backbone model ($\pi_b$), offering (1) transferability across different backbone models or tasks; and (2) composability by combining multiple specialized reasoning modules through reward optimization.

LLMs of different sizes. In contrast, UniR trains a compact reasoning module that interacts with a frozen backbone at the logit level, making training efficient and producing a flexible, architecture-agnostic component for transfer and guidance.

**Guided LLM generation.** Guiding the generation of frozen LLMs at inference offers a flexible way to improve outputs without full retraining (Li et al., 2023). Some plug-and-play approaches attach a separately trained module to a frozen backbone. TART (Bhatia et al., 2023) trains a separate module attached to a frozen base LLM to perform classification. While TART operates at the embedding level, UniR operates at the logit level to guide the base LLM. Recent techniques focus on finer-grained, token-level guidance (Xu et al., 2024; Yang et al., 2023). GenARM (Xu et al., 2024) proposes a token-level reward model, trained on human preference data with DPO (Rafailov et al., 2023), to guide a frozen LLM. While both GenARM and UniR employ a token-level guiding module, GenARM focuses on training the reward model separately using preference data. In contrast, UniR trains a reasoning module to maximize verifiable rewards, avoiding the need for costly preference data collection. In addition, several works (Mudgal et al., 2023; Han et al., 2024) train a value function using temporal difference learning. Instead, UniR directly trains the reasoning module using a policy gradient algorithm, collaborating with a backbone model.

## 3 PROBLEM DEFINITION

A growing trend in enhancing LLM reasoning is Reinforcement Learning with Verifiable Rewards (RLVR), where models are guided by predefined and automatically checkable signals—such as correctness or rule-based verification—rather than relying on human preference data. This approach has been adopted in models such as DeepSeek-R1 (DeepSeek-AI, 2025), and recent RLVR research has demonstrated its effectiveness across diverse domains (Su et al., 2025a; Zhao et al., 2025). Following this direction, we focus on tasks where performance can be assessed reliably via such verifiable rewards. We denote the predefined reward as $r(x, y)$ for a given prompt $x$ from a dataset $\mathcal{D}$ and a generated output sequence $y$. Standard RL fine-tuning aims to find an optimal policy, $\pi_\theta^*(y|x)$, that maximizes the expected reward while remaining close to a backbone policy ($\pi_b(y|x)$), typically a pre-trained base LLM. This objective is formulated as:

$$\max_{\pi_\theta} \mathbb{E}_{x\sim\mathcal{D}, y\sim\pi_\theta(y|x)}[r(x, y)] - \beta \mathbb{D}_{KL}[\pi_\theta(y|x)||\pi_b(y|x)], \tag{1}$$

where $\mathbb{D}_{KL}[\cdot||\cdot]$ represents the Kullback-Leibler divergence, and $\beta > 0$ controls the strength of the regularization towards the backbone policy $\pi_b$. The optimal policy $\pi_\theta^*(y|x)$ for this objective can be expressed in a closed form as (Ziebart et al., 2008; Rafailov et al., 2023):

$$\pi_\theta^*(y|x) = \frac{1}{Z(x)} \pi_b(y|x) \exp\left(\frac{1}{\beta} r(x, y)\right), \tag{2}$$

where $Z(x) = \sum_y \pi_b(y|x) \exp(\frac{1}{\beta} r(x, y))$ is a partition function. A key challenge in directly applying Eq. (2) for autoregressive generation is that the trajectory-level reward $r(x, y)$ can be obtained

only after the full sequence $y$ is complete. However, it does not provide token-specific feedback during the generation process. For fine-grained guidance of an LLM's reasoning steps, per-token signals are more desirable. Theoretically, the optimal token-level decision rule for the objective in Eq. (1) can be expressed using the optimal soft Q-function, $Q^*(a_t|s_t)$ where $s_t = (x, y_{<t})$ is the current state and $a_t = y_t$ is the next token. This $Q^*(a_t|s_t)$ represents the maximum expected future return following the optimal policy $\pi_\theta{}^*$. The optimal token-level policy is then given by (Haarnoja et al., 2017; Ziebart et al., 2008):

$$\log \pi_\theta{}^*(y_t|x, y_{<t}) = \log \pi_b(y_t|x, y_{<t}) + \frac{1}{\beta} Q^*(y_t|x, y_{<t}) - \log Z'(x, y_{<t}), \qquad (3)$$

where $Z'(x, y_{<t})$ is the per-step partition function (Uehara et al., 2025; Haarnoja et al., 2017). However, guiding with $Q^*$ is impractical since it requires knowledge of the optimal policy $\pi_\theta{}^*$ itself. This challenge motivates the need to bridge observable trajectory-level rule-based rewards with a practical formulation that enables effective token-level guidance.

## 4 UniR: Universal Reasoner

To circumvent the direct estimation of $Q^*$, our core idea is to learn a separate, typically smaller, reasoning module, denoted as $\pi_r(y|x)$ with parameters $\phi$. This policy is specifically trained to capture the desired reasoning or task-specific behaviors that lead to high rewards. We then use this learned reasoning module to guide a larger LLM.

### 4.1 Modeling Trajectory Rewards via the Reasoning Module

To connect trajectory-level reward $r(x, y)$ to our reasoning module $\pi_r$ and enable its use in a token-by-token guidance mechanism, we posit that the reward can be represented by the sum of log-probabilities of the reasoning module $\pi_r$:

$$\frac{1}{\beta} r(x, y) = \sum_{t=1}^{|y|} \log \pi_r(y_t|x, y_{<t}; \phi). \qquad (4)$$

Here, $\beta$ is the same temperature parameter from Eq. (1). This formulation implies that sequences $y$ assigned a higher sum of token-level log-probabilities by $\pi_r$ should correspond to sequences achieving higher rewards $r(x, y)$. The feasibility of representing a reward function via the log-likelihood of a policy is supported by theoretical work in (Xu et al., 2024), which claims that reward can be represented using the parameterization $\log \pi_r(y|x)$ for some probability distribution $\pi_r(y|x)$. This assumption is further supported by recent works (Li et al., 2025; Wulfmeier et al., 2024) that language models trained via next-token prediction implicitly learn endogenous reward signals.

Since $\log \pi_r(y|x) = \sum_{t=1}^{|y|} \log \pi_r(y_t|x, y_{<t})$ by the chain rule, this suggests that a sufficiently expressive reasoning module $\pi_r$ can indeed model any trajectory-level reward function $r(x, y)$. By substituting the definition from Eq. (4) into the optimal policy form of Eq. (2), we derive the structure of our guided policy for autoregressive generation of the next token $y_t$. This translates to combining their next-token predictions:

$$\log \pi_\theta(y_t|x, y_{<t}) = \log \pi_b(y_t|x, y_{<t}) + \log \pi_r(y_t|x, y_{<t}) - \log Z'(x, y_{<t}), \qquad (5)$$

where $Z'(x, y_{<t})$ is the per-step partition function. Accordingly, the logits from the frozen backbone $\pi_b$ and the learned module $\pi_r$ can be separable to produce the logits for the guided policy $\pi_\theta$.

This additive and separable guidance mechanism extends to scenarios involving multiple reward criteria. Instead of training a single monolithic model to jointly optimize for all objectives, UniR allows for the training of several distinct reasoning modules $\{\pi_r{}^1, \cdots, \pi_r{}^N\}$, each specialized for a different reward function $r_i(x, y)$. At inference, these specialized modules can be combined by taking a weighted sum of their logits and adding them to the backbone model's logits:

$$\log \pi_\theta(y_t|x, y_{<t}) \propto \log \pi_b(y_t|x, y_{<t}) + \sum_{i=1}^{N} \alpha_i \log \pi_r{}^i(y_t|x, y_{<t}), \qquad (6)$$

where $\alpha_i \geq 0$ controls the influence of each reasoning module $\pi_r{}^i$. This approach offers significant flexibility, as the weights $\alpha_i$ can be adjusted at inference time to explore dynamic optimization across multiple objectives without any retraining. This additive combination of logits is not a heuristic but the principled solution to a standard KL-regularized optimization problem, $\max_{\pi_\theta} \mathbb{E}_{y \sim \pi_\theta}[\sum_i \alpha_i r_i(x,y)] - \mathbb{D}_{KL}(\pi_\theta \| \pi_b)$, a common framework for multi-objective alignment (Abdolmaleki et al., 2020; Dekoninck et al., 2023; Xu et al., 2024). The goal is to find the optimal policy $\pi_\theta(y|x) \propto \pi_b(y|x) \prod_i \exp(\frac{1}{\beta}\alpha_i r_i(x,y))$. By substituting our reward definition from Eq. (4), taking the logarithm of this solution directly yields the linear combination of logits in Eq. (6).

## 4.2 TRAINING $\pi_r$ WITH GRPO

The parameters $\phi$ of the reasoning module $\pi_r$ are trained to maximize the expected predefined reward $r(x,y)$, not the sum of its own log-probabilities defined in Eq. (4). We employ Group Relative Policy Optimization (GRPO) (Shao et al., 2024) for this purpose. For each input $x \sim \mathcal{D}$, we generate a group of $G$ candidate responses $\{y^1, \cdots, y^G\}$ by sampling from $\pi_\theta(y^i|x; \phi_{old})$, where $\phi_{old}$ denotes the parameters of the reasoning module from the previous iteration. For each generated sequence $y^i$, we compute its external reward $r(x, y^i)$ (denoted $r_i$ for brevity) and then calculate the normalized advantage $A_i$:

$$A_i = (r_i - \text{mean}(\{r_1, r_2, \cdots, r_G\})) / \text{std}(\{r_1, r_2, \cdots, r_G\}). \tag{7}$$

In GRPO, the objective function for updating the parameters $\phi$ of the reasoning module is then:

$$\mathcal{L}_{\text{GRPO}}(\phi) = -\mathbb{E}_{x \sim \mathcal{D}, \, y^i \sim \pi_\theta(y^i|x;\phi_{old})} \left[ \frac{1}{G} \sum_{i=1}^{G} \left\{ \min\left( \frac{\pi_\theta(y^i|x)}{\pi_\theta(y^i|x; \phi_{old})} A_i, \right. \right. \right.$$

$$\left. \left. \left. \text{clip}\left( \frac{\pi_\theta(y^i|x)}{\pi_\theta(y^i|x; \phi_{old})}, \, 1-\epsilon, \, 1+\epsilon \right) A_i \right) - \beta_{KL} \mathbb{D}_{KL}[\pi_\theta(y^i|x; \phi) || \pi_\theta(y^i|x; \phi_0)] \right\} \right]. \tag{8}$$

Here, $\epsilon$ is the clipping hyperparameter from PPO-clip, $\phi_0$ is initial value of $\phi$, and $\beta_{KL}$ controls the KL-regularization term. The learning direction is determined by the ratio $\frac{\pi_\theta(y^i|x,\phi)}{\pi_\theta(y^i|x;\phi_{old})}$, which simplifies to $\frac{\pi_r(y^i|x,\phi)}{\pi_r(y^i|x;\phi_{old})}$ as the terms from the $\pi_b$ cancel out, thereby the gradient $\nabla_\phi \mathcal{L}_{\text{GRPO}}$ only affects the parameters of $\pi_r$ since $\pi_b$ is frozen.

## 4.3 INTERPRETING $\pi_r$ WITH $Q^*$

We now provide theoretical grounding to understand what the learned reasoning module $\pi_r$ captures at each token-level decision point. This provides a theoretical justification for interpreting $\log \pi_r$ as a meaningful guidance signal that drives the policy towards optimality for the trajectory-level objective.

**Theorem 1.** *(Informal) If the reasoning module $\pi_r$ satisfies Eq. (4), such that its summed per-token log-probabilities accurately approximate the scaled reward for trajectories $y$ that are consistent with the optimal policy $\pi_\theta{}^*$, then*

$$\log \pi_r(y_t|x, y_{<t}) = \frac{1}{\beta} Q^*(y_t|x, y_{<t}). \tag{9}$$

*Proof.* We assume guided policy $\pi_\theta$ has converged to the optimal policy $\pi_\theta{}^*$ for all states $s_t = (x, y_{<t})$ and actions $a_t = y_t$ by optimizing the GRPO objective to maximize the reward $r(x,y)$ since the KL-regularized objective typically admits a unique optimum. Given that the trajectory-level policies are identical, their per-token conditional distributions must also match for any $s_t$. Comparing Eq. (5) with the standard form of the optimal policy expressed in terms of the soft Q-function (Eq. (3)) shows that $\log \pi_r(y_t \mid x, y_{<t})$ corresponds to the scaled optimal soft Q-value. $\square$

# 5 EXPERIMENTS

## 5.1 EXPERIMENTAL SETTINGS

We evaluate UniR on two tasks: mathematical problem-solving and machine translation. Mathematical problem-solving requires explicit, step-by-step reasoning, while machine translation involves

Table 1: Zero-shot pass@1 performance with standard deviation on mathematic benchmarks. The out-of-distribution results are obtained using models trained on the MATH training dataset. Models marked with * are taken from the report (Yang et al., 2024b).

| | Method | Trained Model | In-distribution | | Out-of-distribution | | | Avg. |
|---|---|---|---|---|---|---|---|---|
| | | | GSM8K | MATH-500 | AIME24 | Minerva | OlympiadBench | |
| Other 7B | Mathstral-7B-v0.1* | 7B | 84.9 | 56.6 | - | 16.2 | 21.5 | - |
| | Internlm2-math-plus-7B* | 7B | 84.0 | 54.4 | - | 17.3 | 18.8 | - |
| | NuminaMath-7B-CoT* | 7B | 75.4 | 55.2 | - | 19.1 | 19.9 | - |
| | DeepSeekMath-7B-RL* | 7B | 88.2 | 52.4 | - | 20.6 | 19.0 | - |
| Llama3.2-3B | Backbone only | - | $65.6 \pm 1.3$ | $33.0 \pm 2.3$ | $3.7 \pm 1.8$ | $12.5 \pm 2.3$ | $8.5 \pm 2.4$ | 24.7 |
| | Backbone + 1B | - | $63.8 \pm 0.7$ | $35.7 \pm 2.0$ | $5.0 \pm 2.7$ | $13.1 \pm 1.7$ | $9.9 \pm 0.9$ | 25.5 |
| | GRPO LoRA | 3B | $65.8 \pm 0.9$ | $32.1 \pm 1.6$ | $6.0 \pm 2.5$ | $12.0 \pm 1.3$ | $11.9 \pm 0.7$ | 25.6 |
| | GRPO Full | 3B | $65.1 \pm 1.1$ | $34.6 \pm 1.9$ | $4.0 \pm 3.6$ | $12.6 \pm 1.7$ | $10.3 \pm 0.6$ | 25.3 |
| | **Ours** | 1B | $\mathbf{77.5} \pm 1.2$ | $\mathbf{48.8} \pm 1.2$ | $\mathbf{7.3} \pm 2.9$ | $\mathbf{16.6} \pm 1.6$ | $\mathbf{16.4} \pm 0.5$ | **33.3** |
| Qwen2.5-3B | Backbone only | - | $74.4 \pm 1.0$ | $44.2 \pm 1.3$ | $6.3 \pm 3.7$ | $19.2 \pm 1.7$ | $24.6 \pm 0.9$ | 33.7 |
| | Backbone + 1.5B | - | $75.6 \pm 0.9$ | $50.6 \pm 1.6$ | $4.0 \pm 2.1$ | $20.7 \pm 2.3$ | $25.1 \pm 0.9$ | 35.2 |
| | GRPO LoRA | 3B | $75.0 \pm 0.9$ | $60.0 \pm 1.2$ | $5.7 \pm 3.2$ | $24.7 \pm 2.1$ | $27.1 \pm 0.6$ | 38.5 |
| | GRPO Full | 3B | $81.3 \pm 0.5$ | $62.9 \pm 1.3$ | $4.7 \pm 2.3$ | $25.0 \pm 1.0$ | $27.5 \pm 1.3$ | 40.3 |
| | **Ours** | 1.5B | $\mathbf{84.5} \pm 0.7$ | $\mathbf{66.8} \pm 1.4$ | $\mathbf{7.7} \pm 2.7$ | $\mathbf{26.3} \pm 2.0$ | $\mathbf{28.2} \pm 0.8$ | **42.7** |

Table 2: Pass@1 performance with standard deviation on machine translation benchmarks for both English-to-German (EN→DE) and German-to-English (DE→EN) directions. GPT-4.1-nano and Qwen2.5-3B-Instruct were evaluated with the same system and user prompts as used for our method.

| | Method | Trained Model | EN → DE | | | | DE → EN | | | |
|---|---|---|---|---|---|---|---|---|---|---|
| | | | BLEU | CometKiwi | XComet | Avg. | BLEU | CometKiwi | XComet | Avg. |
| Others | Google translator | - | 32.39 | 80.20 | 92.00 | 68.20 | 32.95 | 81.00 | 92.90 | 68.95 |
| | GPT4.1-nano | - | 31.89 | 83.90 | 96.00 | 70.60 | 37.14 | 83.70 | 97.00 | 72.61 |
| | Qwen2.5-3B-Instruct | - | 23.28 | 78.25 | 91.26 | 64.26 | 32.75 | 82.66 | 95.70 | 70.37 |
| Llama3.2-3B | Backbone only | - | $23.19 \pm 0.12$ | $77.72 \pm 0.12$ | $90.97 \pm 0.12$ | 63.96 | $30.65 \pm 0.16$ | $81.27 \pm 0.05$ | $94.53 \pm 0.05$ | 68.82 |
| | Backbone + 1B | - | $24.10 \pm 0.04$ | $78.70 \pm 0.05$ | $91.47 \pm 0.05$ | 64.76 | $31.82 \pm 0.08$ | $81.75 \pm 0.02$ | $94.74 \pm 0.03$ | 69.44 |
| | GRPO LoRA | 3B | $24.18 \pm 0.08$ | $78.93 \pm 0.05$ | $91.97 \pm 0.05$ | 65.03 | $31.39 \pm 0.07$ | $81.73 \pm 0.03$ | $94.95 \pm 0.03$ | 69.36 |
| | GRPO Full | 3B | $25.26 \pm 0.07$ | $80.13 \pm 0.06$ | $92.88 \pm 0.06$ | 66.09 | $35.52 \pm 0.13$ | $80.43 \pm 0.27$ | $92.35 \pm 0.41$ | 69.43 |
| | **Ours** | 1B | $\mathbf{27.03} \pm 0.07$ | $\mathbf{81.43} \pm 0.02$ | $\mathbf{93.70} \pm 0.03$ | **67.39** | $\mathbf{37.21} \pm 0.08$ | $\mathbf{82.82} \pm 0.02$ | $\mathbf{95.89} \pm 0.03$ | **71.97** |

implicit reasoning based on domain-specific knowledge and semantic understanding. These tasks provide complementary benchmarks for assessing our decomposed reasoning module.

**Models and Baselines.** Our experiments utilize models from Qwen2.5 (Yang et al., 2024a) and Llama3.2 (Grattafiori et al., 2024). We train the reasoning module $\pi_r$ alongside a 3B-scale backbone $\pi_b$ in two settings: (1) $\pi_b$ as Qwen2.5-3B with $\pi_r$ as Qwen2.5-1.5B, and (2) $\pi_b$ as Llama3.2-3B with $\pi_r$ as Llama3.2-1B. For the transferability experiment, we use Qwen2.5-14B and Llama3.1-8B as backbone models. Unless stated otherwise, all models are Instruct variants. For baselines, we apply GRPO to fine-tune the entire backbone model (e.g., the 3B-scale $\pi_b$ model) directly on each task using the same predefined rewards. Baseline tuning is implemented through either full model tuning or LoRA-based fine-tuning.

**Dataset and Reward Formulation.** For mathematical reasoning tasks, we use the Math-12k (Lightman et al., 2023; Liu et al., 2025b) and GSM8k (Cobbe et al., 2021) datasets. The reward $r(x, y)$ is determined by the rule-based correctness of the final answer, following the methodology in Dr.GRPO (Liu et al., 2025b). For machine translation, we use the IWSLT 2017 dataset (Cettolo et al., 2017) for English-to-German and German-to-English translation, employing BLEU (Papineni et al., 2002) and CometKiwi (Rei et al., 2022) as reward during training. Following MT-R1-Zero (Feng et al., 2025), we combine lexical and semantic metrics for balanced training.

**Evaluation Metrics.** We evaluate mathematical reasoning using pass@1 accuracy on five benchmarks: GSM8k, MATH-500 (Lightman et al., 2023), AIME24 [1], Minerva (Lewkowycz et al., 2022), and OlympiadBench (He et al., 2024). GSM8k is tested with a model trained by its own training split; the others are evaluated with a model trained by Math-12K. For the translation task, we report BLEU, CometKiwi, and XComet-XL (Guerreiro et al., 2024) scores. We set the temperature to 0.6 and evaluated performance across 10 different seeds. Further details on training hyperparameters, optimization settings, evaluation details are provided in Appendix B, and prompt templates are presented in Appendix E.

---

[1] https://huggingface.co/datasets/AI-MO/aimo-validation-aime

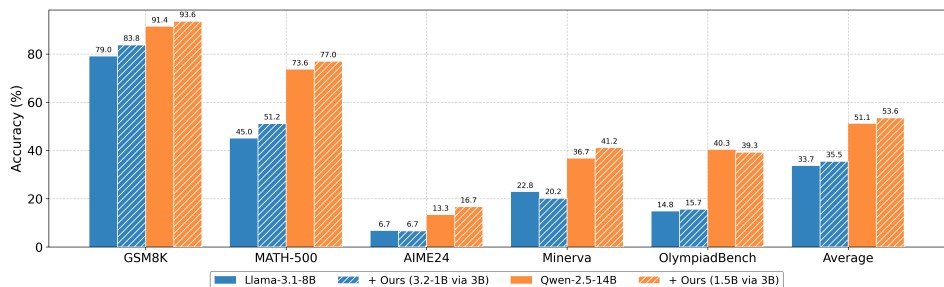

Figure 2: **Effectiveness of Reasoning module Transfer**. Results demonstrate that a trained reasoning module can improve performance when integrated with larger backbone models across diverse mathematical reasoning tasks.

## 5.2 ENHANCING REASONING ABILITY

As shown in Table 1, our framework consistently achieves substantial improvements over GRPO baselines across both in-distribution and out-of-distribution benchmarks. Specifically, UniR achieves an average score of 33.3 and 42.7, surpassing GRPO Full (25.3, 40.3) and GRPO LoRA (25.6, 38.5) with both model families, respectively. In machine translation, as shown in Table 2, UniR also outperforms both GRPO baselines in both translation directions. For EN→DE, UniR improves the BLEU score by +1.77 over GRPO Full, while for DE→EN it improves BLEU by +1.69. Beyond BLEU, semantic metrics such as CometKiwi and XComet also exhibit significant improvements. These results indicate that UniR effectively generalizes beyond explicit reasoning in solving mathematical problems, extending its efficacy to implicit reasoning tasks such as translation.

To isolate the contribution of our reward-driven alignment, we compare UniR against ablative baselines (*e.g.*, Backbone + 1B). This represents a naive ensemble where the logits of the frozen backbone are directly summed with those of a pre-trained reasoning module without any of the reward optimization proposed by UniR. As shown in Tables 1 and 2, merely combining pre-trained model outputs performs worse than the UniR method. This highlights the need for our reward alignment process, which is crucial for transforming the reasoning module into an effective guide.

## 5.3 TRANSFERABILITY AND COMPOSABILITY

**Transferability to Larger Backbone LLMs.** Figure 2 illustrates the transferability of our reasoning module $\pi_r$, originally trained on Math-12k and GSM8k datasets with a 3B-scale backbone. We evaluate its effectiveness when deployed to guide larger frozen backbone models with 8B and 14B parameters. Across most benchmarks, the transferred $\pi_r$ improves average reasoning performance (35.5 and 53.6) compared to the respective backbone baselines (33.7 and 51.1), even without additional fine-tuning. This suggests that $\pi_r$ is not narrowly specialized to its original backbone but instead functions as a generalizable reasoning augmentation module to enhance the reasoning capabilities of stronger models at inference time in a resource-efficient framework.

**Combining Specialized Reasoning Modules.** To investigate the advantages of integrating multiple specialized reasoning capabilities enabled by UniR, we design a composite task: translating German math problems from the MGSM dataset (Shi et al., 2022) into English. We utilize two distinct reasoning modules: $\pi_r^{\text{Math}}$, a module specialized for mathematical problem-solving on the GSM8k dataset, and $\pi_r^{\text{Trans}}$, a module trained for German-to-English translation. These modules are combined with a frozen base LLM ($\pi_b$) through a weighted summation of their logits:

$$\log \pi_b + \alpha \log \pi_r^{\text{Math}} + (1 - \alpha) \log \pi_r^{\text{Trans}}, \quad (10)$$

where $\alpha \in [0, 1]$ is a coefficient that balances the influence between both modules. We employ GPT-4.1-nano to evaluate translation quality and the accuracy of the generated output. Detailed configurations for this experiment are provided in Appendix E.3.

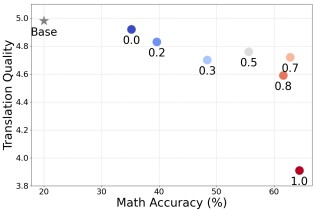

Figure 3: **Performance on a German-to-English Math problem-solving task.** The numbers in the figure indicate the value of $\alpha$ and "Base" denotes backbone-only method.

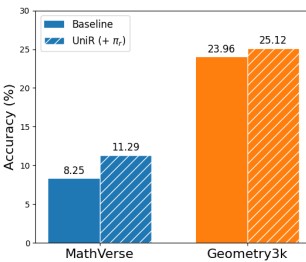

Figure 4: UniR Performance on Visual Math Reasoning.

Table 3: Evaluation of UniR on ICU readmission, length of stay (LOS), and mortality prediction.

| Task | Method | Reasoning module | Acc. | F1 |
|---|---|---|---|---|
| Readmission | Backbone only | - | 39.46 | 25.87 |
| | UniR | Readmission | **52.91** | 26.77 |
| | UniR | Readmission + LOS | 44.88 | **28.79** |
| Length of stay (LOS) | Backbone only | - | 9.64 | 5.92 |
| | UniR | LOS | **67.77** | 16.25 |
| | UniR | Readmission + LOS | 61.75 | **18.32** |
| Mortality | Backbone only | - | 39.76 | 28.27 |
| | UniR | Readmission + LOS | **63.65** | **35.18** |

As depicted in Figure 3, the backbone-only model (Base) achieves a high translation quality but yields poor math accuracy since it only perform translation. When the translation module is added ($\alpha = 0$), math accuracy improves even though the module was trained only for translation. This phenomenon highlights that the $\pi_r^{\text{Trans}}$ modules' guidance steers the backbone reasoning states into a English-understanding mode. Once the backbone is controlled to think in coherent English, its own latent mathematical ability suppressed by the German input is enhanced, causing the accuracy jump. Beyond this baseline clarification, we observe that increasing the weight $\alpha$ generally leads to improved accuracy in solving the mathematical problems, while a higher weight for the translation module tends to improve translation quality. This highlights a crucial capability of UniR: the ability to combine and balance multiple specialized reasoning modules. This modularity allows for precise control over the policy model's behavior, leading to improved performance on multi-faceted tasks. We illustrate the example of the responses in Figure 18.

## 5.4 ADDITIONAL EXPERIMENTS: EXTENSION AND ANALYSIS

**Extension to Vision Language Models.** To investigate whether our UniR framework can transfer reasoning ability in multimodal contexts, we extend it by employing a Vision Language Model (VLM) as the backbone. Specifically, we utilize Qwen2.5-VL-3B-Instruct (Bai et al., 2025) as the frozen backbone model $\pi_b$. This VLM is guided by a 0.5B Qwen-based reasoning module $\pi_r$, which is trained on the text-based GSM8k dataset. For evaluation, we use the "Vision Intensive" questions in MathVerse (Zhang et al., 2024) and Geometry3k (Lu et al., 2021) benchmarks, which feature mathematical problems accompanied by visual diagrams, thus requiring both visual understanding and mathematical reasoning. Evaluation protocols and system prompts follow those used in the GSM8k experiments. An example is depicted in Figure 20.

As shown in Figure 4, UniR significantly improves performance over the baseline VLM on both benchmarks. These enhancements are particularly noteworthy because the reasoning module $\pi_r^{Math}$ was trained exclusively on text-based mathematical problems, with no access to visual information. This aligns with prior work (Liu et al., 2025a), which shows that text-only post-training can improve multimodal reasoning capabilities. Unlike monolithic post-training, UniR achieves this through a plug-and-play module that guides frozen VLMs, providing a flexible and efficient way to extend reasoning to multimodal domains.

**Generalization to Medical Domain.** To further assess the generalizability of our method, we conduct additional experiments in the medical domain using the MIMIC-IV dataset (Johnson et al., 2020). Access to the dataset was obtained through PhysioNet (Goldberger et al., 2000) under the appropriate data use agreements. We evaluate UniR on two primary tasks: (1) Readmission Prediction and (2) Length of Stay (LOS) Prediction. In the readmission prediction task, the model predicts whether a patient will be readmitted within 15 days of discharge (binary classification). For LOS prediction, we train UniR to categorize the patient's ICU stay length into four classes: short stay ($\leq$ 1 day), moderate stay (2–7 days), extended stay (8–14 days), and long-term stay ($>$ 14 days). Detailed settings are provided in Appendix B.3.1.

We use Qwen2.5-3B as the backbone model and Qwen2.5-0.5B as reasoning modules. Table 3 summarizes the results. For the readmission task, UniR improves accuracy by 13.45 points and F1-score by 0.9 points over the baseline. For the LOS task, UniR achieves an accuracy of 67.77% and an F1-score of 16.25%, representing a dramatic improvement over the baseline of 9.64% and 5.92%, respectively. Moreover, we evaluate composability of our framework in the medical setting. While this composition decreases accuracy compared to using each module individually, it consistently

Table 4: Comparison on math problems in Japanese (JP) and Spanish (ES).

| Method | Accuracy (%) | | Translation Quality ↑ | |
|---|---|---|---|---|
| | JP | ES | JP | ES |
| Backbone | 42.0 | 62.4 | 4.63 | 4.98 |
| Ours | 50.4 | 66.8 | 4.64 | 4.75 |

Table 5: Evaluation of using a small pre-trained reasoning model on math reasoning within the UniR framework.

| Backbone Model $\pi_b$ | Reasoning Module $\pi_r$ | Train with UniR | Avg. |
|---|---|---|---|
| DeepSeek 1.5B | - | X | 32.8 |
| Qwen2.5 3B | DeepSeek 1.5B | X | 43.7 |
| Qwen2.5 3B | DeepSeek 1.5B | O | 46.9 |

Table 6: Comparison of inference speed with UniR acceleration.

| Method | Sec./Token ↓ |
|---|---|
| Backbone only | 0.0262 |
| UniR (naive) | 0.0414 |
| UniR + CoS | 0.0279 |

improves F1-scores. Furthermore, the combined inference generalizes to the unseen mortality prediction task, which is designed for predicting mortality in binary classification, yielding substantial gains over the baseline without task-specific training. These results suggest that complementary knowledge from different tasks can be integrated, which is beneficial in medical reasoning as their combination provides a richer basis for risk estimation, extending even to mortality. The example is shown in Figure 21.

**Generalization across Low-resource Languages.** We investigate whether the UniR math reasoning module $\pi_r^{Math}$, trained solely on English math problems, can generalize to low-resource languages, Japanese and Spanish, which are underrepresented in pre-training corpora. Each task is tested on MGSM, with accuracy and translation quality, which measure fluency and consistency in each language using the GPT-4.1-nano model. As shown in Table 4, UniR consistently improves math reasoning accuracy even when both inputs and outputs are entirely in low-resource languages, highlighting its generalization capability beyond English. See Appendix B.3.2 for details.

**Using Pre-trained Reasoning Model Directly as Guidance.** We investigate the effectiveness of using a pre-trained small reasoning model as the reasoning module $\pi_r$ within the UniR framework. Specifically, we use DeepSeek-R1-Distill-Qwen-1.5B (DeepSeek-AI, 2025), a distilled 1.5B model trained via supervised fine-tuning on reasoning data. We evaluate its performance with and without further UniR optimization. Table 5 as an average across five mathematical benchmarks (see Table 12 for the full breakdown), shows that integrating a pre-trained reasoning model into UniR improves performance in mathematical reasoning, even without additional training. We believe this is due to the theoretical alignment between Thm. 1 and recent findings (Wulfmeier et al., 2024; Li et al., 2025) which show that models trained via next-token prediction implicitly learn soft Q-functions and encode endogenous reward signals. Further fine-tuning through UniR enhances reasoning capabilities beyond the simple combination. This highlights the potential of reusing pre-trained small reasoning models within UniR fine-tuning for improving reasoning performance.

**Enhancing Inference Efficiency with Speculative Decoding.** Integrating additional reasoning modules can increase per-token inference time. As shown in Table 6, integrating UniR naively increases per-token inference time compared to the backbone alone. To mitigate this, we integrate UniR with a recent speculative decoding method, specifically CoS (Fu et al., 2025), which enables collaborative inference by leveraging multiple model logits in parallel. This approach effectively reduces the overhead, bringing inference speed close to that of the backbone model. See Appendix B.3.3 for detailed experimental setup.

**Analysis of Reward Decomposition.** To empirically validate our reward modeling (Eq. 4), we analyze the summed log-probabilities ($\sum_t \log \pi_r$) in generated responses on the GSM8k test set. As detailed in Appendix C.1 and shown in its accompanying figure, our analysis reveals a strong positive correlation: correct responses ($r = 1$) are assigned significantly higher log-probabilities than incorrect ones ($r = 0$). This effect is amplified by our training method. While the pre-trained module exhibits moderate separation, the distinction becomes far more pronounced after UniR training, confirming that our framework effectively trains the module to internalize the reward signal.

## 6 CONCLUSION

In this paper, we present Universal Reasoner, a novel plug-and-play reasoning module that directly optimizes predefined rewards by focusing solely on a compact reasoning module while keeping the larger backbone model frozen, enabling efficient training. Our approach exhibits strong transferability; the reasoning module can effectively guide backbone models of various sizes without requiring further fine-tuning. Furthermore, UniR facilitates the combination of modules specialized for different tasks. UniR demonstrates generalization not only across modalities but also across different

domains, highlighting its potential to integrate multiple domain-specific modules and achieve synergistic improvements across diverse tasks.

**Limitations.** While UniR is designed to be architecture-agnostic, its application requires tokenizer alignment. The performance of our framework is inherently bounded by the capabilities of the frozen backbone model $\pi_b$, as $\pi_r$ primarily guides its existing representations. While additive composition is grounded in multi-objective optimization, the potential for negative interference between modules with conflicting reward signals remains an area for future theoretical investigation.

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

## A   THEORETICAL JUSTIFICATION

### A.1   REWARD PARAMETERIZATION

To justify the reward parameterization in Eq. (4), which decomposes the trajectory-level reward function into a token-level sum, we first formally prove that this parameterization is equivalent and expressive for any reward function within our optimization objective.

**Lemma 2** (Theorem 3 in Xu et al. (2024))**.** *For any trajectory-level reward function $r(x, y)$ and any $\beta > 0$, there exists an equivalent reward function $\hat{r}(x, y)$ and a probability distribution $\pi_r(y|x)$ such that $\hat{r}(x, y) = \beta \log \pi_r(y|x)$, where $r(x, y)$ and $\hat{r}(x, y)$ induce the identical optimal policy $\pi_\theta{}^*$ for the KL-regularized objective in Eq. (1).*

*Proof.* As shown in Eq. (2), the unique optimal policy $\pi_\theta{}^*$ for the KL-regularized objective has the closed-form solution. Following Theorem 3 in Xu et al. (2024), we define two reward functions $r(x, y)$ and $\hat{r}(x, y)$ as equivalent if they induce the same optimal policy. This holds if they differ only by a function $f(x)$ that is independent of the trajectory $y$:

$$\hat{r}(x, y) = r(x, y) - f(x). \tag{11}$$

This is because the $f(x)$ term is absorbed into the partition function $Z(x)$.

Our goal is to find a function $f(x)$ that allows $\hat{r}(x, y)$ to be written in the form $\beta \log \pi_r(y|x)$ for some valid probability distribution $\pi_r(y|x)$. Let us define a probability distribution $\pi_r(y|x)$ using the given reward function $r(x, y)$:

$$\pi_r(y|x) := \frac{\exp(r(x, y)/\beta)}{\sum_{y'} \exp(r(x, y')/\beta)} \tag{12}$$

By construction, $\sum_y \pi_r(y|x) = 1$, so $\pi_r$ is a valid probability distribution.

Let us define our new reward function $\hat{r}(x, y)$ using this $\pi_r(y|x)$, as required by the lemma:

$$\hat{r}(x, y) := \beta \log \pi_r(y|x). \tag{13}$$

Substitute the definition of $\pi_r(y|x)$ into this equation:

$$\hat{r}(x, y) = \beta \log \left( \frac{\exp(r(x, y)/\beta)}{\sum_{y'} \exp(r(x, y')/\beta)} \right) \tag{14}$$

$$= \beta \left[ \log \left( \exp \left( \frac{r(x, y)}{\beta} \right) \right) - \log \left( \sum_{y'} \exp \left( \frac{r(x, y')}{\beta} \right) \right) \right] \tag{15}$$

$$= r(x, y) - \beta \log \left( \sum_{y'} \exp \left( \frac{r(x, y')}{\beta} \right) \right) \tag{16}$$

$$= r(x, y) - f(x), \tag{17}$$

where $f(x) = \beta \log \left( \sum_{y'} \exp(r(x, y')/\beta) \right)$. Therefore, $\hat{r}(x, y)$ and $r(x, y)$ are equivalent, as they induce the identical optimal policy $\pi_\theta^*$. This proves that any reward function $r(x, y)$ has an equivalent counterpart $\hat{r}(x, y)$ that can be represented as a scaled log-probability of a distribution $\pi_r$.

$\square$

**Corollary 2.1.** *The trajectory-level reward $\hat{r}(x, y)$ proven in Lemma 2 can be decomposed into a sum of token-level log-probabilities.*

*Proof.* By the chain rule of probability, trajectory-level distribution $\pi_r(y|x)$ can be decomposed as:

$$\log \pi_r(y|x) = \sum_{t=1}^{|y|} \log \pi_r(y_t|x, y_{<t}). \tag{18}$$

Substituting this into the result from Lemma 2:

$$\hat{r}(x,y) = \beta \log \pi_r(y|x) = \beta \sum_{t=1}^{|y|} \log \pi_r(y_t|x, y_{<t}). \tag{19}$$

Dividing by $\beta$, we arrive at the form used in our paper Eq. (4):

$$\frac{1}{\beta}\hat{r}(x,y) = \sum_{t=1}^{|y|} \log \pi_r(y_t|x, y_{<t}). \tag{20}$$

This formally justifies that our parameterization in Eq. (4) is not a limiting assumption, but a fully general way to represent any reward function for our objective. $\square$

### A.2 Interpretation of $\pi_r$ at the Optimum

In this section, we derive the token-level structure of the optimal policy $\pi_\theta^*$ and use it to motivate the design of our guided policy $\pi_\theta$.

From Lemma 2 and Corollary 2.1, we substitute the equivalent reward $r(x,y)$ (omitting the $\hat{\cdot}$ for clarity) its token-decomposed log-probability from Eq. (2):

$$\pi_\theta^*(y|x) \propto \pi_b(y|x) \exp\left( \sum_{t=1}^{|y|} \log \pi_r(y_t|x, y_{<t}) \right) \tag{21}$$

$$= \pi_b(y|x) \prod_{t=1}^{|y|} \pi_r(y_t|x, y_{<t}) \tag{22}$$

$$= \prod_{t=1}^{|y|} \pi_b(y_t|x, y_{<t}) \pi_r(y_t|x, y_{<t}) \tag{23}$$

By definition, the trajectory-level policy $\pi_\theta^*(y|x)$ is also a product of its own token-level conditional policies: $\pi_\theta^*(y|x) = \prod_t \pi_\theta^*(y_t|x, y_{<t})$. By matching the terms inside the product for each step $t$, we derive the token-level optimal policy defined in Eq. (3).

Motivated by this derivation, we define our trainable policy $\pi_\theta$ to have this exact mathematical structure, but replacing the ideal $\pi_r$ with our learnable reasoning module $\pi_r(\cdot; \phi)$ in Eq. (5).

**Theorem 3.** *Let $\pi_\theta^*$ be the unique optimal policy for the KL-regularized objective Eq. (1) with reward $r(x,y)$. The token-level distribution of $\pi_\theta^*$ is given by Eq. (3):*

$$\log \pi_\theta^*(y_t|x, y_{<t}) = \log \pi_b(y_t|x, y_{<t}) + \frac{1}{\beta}Q^*(y_t|x, y_{<t}) - \log Z'(x, y_{<t}), \tag{24}$$

*where $Q^*(y_t|x, y_{<t})$ is the optimal soft Q-function for the reward $r(x,y)$ and $Z'$ is the per-step partition function. As the KL-regularized objective drives $\pi_\theta \to \pi_\theta^*$, we can formally interpret the trained reasoning module's output as the scaled optimal soft Q-function:*

$$\log \pi_r(y_t|x, y_{<t}) = \frac{1}{\beta}Q^*(y_t|x, y_{<t}) \tag{25}$$

*Proof.* The objective in Eq. (1) has a unique optimal solution $\pi_\theta^*$. Let us assume the optimization is successful and $\pi_\theta$ has converged to $\pi_\theta^*$:

$$\log \pi_\theta(y_t|x, y_{<t}; \phi) = \log \pi_\theta^*(y_t|x, y_{<t}). \tag{26}$$

Now, substitute the definitions from Eq. (3) and Eq. (5) into this equality. Then, we get:

$$\log \pi_r(y_t|x, y_{<t}; \phi) = \frac{1}{\beta}Q^*(y_t|x, y_{<t}). \tag{27}$$

This proves that the our reasoning module $\pi_r(\cdot; \phi)$ is learning to approximate the scaled optimal soft Q-function via the KL-regularized optimization. $\square$

# B EXPERIMENTAL DETAILS

## B.1 HYPERPARAMETER SETUP

The detailed hyperparameter settings are provided in Table 7. For the fair comparison, we used the same hyperparameters across all methods, including the baseline model, GRPO with full fine-tuning, GRPO with LoRA, and simple logit-sum. Specifically, we set the KL divergence coefficient $\beta = 0.04$ for GRPO, following the default value in the Hugging Face TRL library [2]. However, in our UniR framework, the reasoning module $\pi_r$ is often initialized from a general pre-trained small model, which may not possess strong task-specific reasoning capabilities. Regularizing $\pi_\theta$ towards $\pi_b$ which is heavily influenced by the initial, potentially $\pi_{r_{\text{old}}}$ can impede the learning of high-reward reasoning paths. Therefore, we omit such an explicit KL regularization by $\beta_{KL} = 0$, allowing $\pi_r$ more freedom to explore and learn effective reasoning strategies driven by the predefined rewards. This choice aligns with findings in related works such as DAPO (Yu et al., 2025), which also found benefits in adjusting or omitting certain regularization components.

Among the reward configurations, *rule_based_accuracy* assigns a reward of 1 if the numeric value between the <answer> and </answer> tags matches the ground truth and 0 otherwise. The *boxed_reward* evaluates the value enclosed within the \boxed{} format, assigning a reward of 1 if it matches the correct answer and 0 otherwise. Similarly the *tag_based_reward* evaluates the value enclosed within the answer tag. For the implementation of *boxed_reward* and *tag_based_reward* we adopted the official code from the DR.GRPO repository (Liu et al., 2025b)[3]. For BLEU, we used the default implementation of BLEU scoring provided by SacreBLEU[4].

Table 7: Hyperparameter setups.

| Parameter | GSM8K | Math-12K-Qwen | Math-12K-Llama | Translation |
|---|---|---|---|---|
| *Training Configuration* | | | | |
| gradient_checkpointing | true | true | true | true |
| gradient_checkpointing_kwargs | use_reentrant: false | use_reentrant: false | use_reentrant: false | use_reentrant: false |
| learning_rate | 1.0e-06 | 1.0e-06 | 1.0e-06 | 1.0e-06 |
| lr_scheduler_type | cosine with min_lr: 0.1 | cosine with min_lr: 0.1 | cosine with min_lr: 0.1 | cosine with min_lr: 0.1 |
| warmup_ratio | 0.1 | 0.1 | 0.1 | 0.1 |
| max_steps | 1000 | 1000 | 1000 | 10000 |
| effective_batch_size | 64 | 32 | 32 | 32 |
| max_prompt_length | 512 | 512 | 512 | 512 |
| max_completion_length | 1024 | 2048 | 2048 | 1024 |
| num_generations | 8 | 8 | 8 | 8 |
| temperature | 0.7 | 0.7 | 0.7 | 0.7 |
| $\alpha$ | 1.0 | 1.0 | 1.0 | 1.0 |
| *Evaluation Settings* | | | | |
| max_prompt_length | 512 | 512 | 512 | 512 |
| max_completion_length | 1024 | 2048 | 2048 | 250 |
| temperature | 0.6 | 0.6 | 0.6 | 0.6 |
| $\alpha$ | 1.0 | 1.0 | 1.0 | 1.0 |
| *Reward Configuration* | | | | |
| reward_funcs | rule_based_accuracy | boxed_reward | tag_based_reward | BLEU, cometkiwi |
| reward_weights | 1.0 | 1.0 | 1.0 | 1.0, 1.0 |

## B.2 EVALUATION DETAIL

For evaluation, BLEU and accuracy were computed using the same reward logic applied during training for each dataset. Translation baselines included outputs from Google Translator, implemented via the `GoogleTranslator` interface of the `deep-translator` library[5]. GPT-4.1-nano was also used for comparison, with decoding performed at a sampling temperature of 0.7. We employ CometKiwi [6] and XComet-XL [7] scores to evaluate translation quality. For the main results in Table 1, 2, and 9, decoding was performed with a sampling temperature of 0.6. We report the

---

[2] https://github.com/huggingface/trl
[3] https://github.com/sail-sg/understand-r1-zero
[4] https://github.com/mjpost/sacrebleu
[5] https://github.com/nidhaloff/deep-translator
[6] https://huggingface.co/Unbabel/XCOMET-XL
[7] https://huggingface.co/Unbabel/wmt22-CometKiwi-da

mean and standard deviation over ten runs with different seeds (1-10). All other experiments were conducted deterministically using greedy decoding `do_sample=False`.

### B.3 ADDITIONAL EXPERIMENTAL SETUP

#### B.3.1 MEDICAL REASONING

Table 8: Hyperparameter setup for training readmission and length of stay tasks.

| Parameter | Value |
|---|---|
| *Training Configuration* | |
| max_steps | 1000 |
| max_prompt_length | 3072 |
| max_completion_length | 1024 |
| num_generations | 4 |
| *Reward Configuration* | |
| reward_funcs | accuracy |
| reward_weights | 1.0 |

The detailed hyperparameter settings are summarized in Table 8. A key difference from previous tasks was the use of longer prompt and completion lengths to accommodate the more extensive data in the medical domain. We provide the prompt in Figure 15. The patient information contains a Patient ID and information on conditions, procedures, and medications for each visit. When composing the two task modules, we set the weight for each module to $\alpha_i = 1$, as specified in Eq. ( 6). All other prompts and parameters remain the same as when a single module was used. For label construction, we map all categorical values to numeric labels. In the readmission task, patients readmitted within 15 days are labeled 1, while others are labeled 0. For the length of stay prediction task, we use four categories: 0 ($\leq$ 1 day), 1 (2–7 days), 2 (8–14 days), and 3 ($>$ 14 days). Finally, in the mortality prediction task, patients who died are labeled 1, and survivors are labeled 0. For reward design, the model receives a reward of 1 if the numeric value extracted between the <answer> and </answer> tags matched the ground truth, and 0 otherwise.

#### B.3.2 LOW-RESOURCE LANGUAGE

We follow the experimental setting described in Appendix E.3. For prompts, we adapt those shown in Figure 13 and Figure 14 by replacing the language tokens 'German' and 'English' with low-resource languages such as 'Japanese' and 'Spanish'. For each language, we test each model on math problems written and answered in the original language using MGSM dataset.

#### B.3.3 INFERENCE WITH SPECULATIVE DECODING

Collaborative decoding is a technique that improves generation by combining outputs, such as probability distributions or logits, from multiple models at each step. Building on Speculative Decoding (Leviathan et al., 2023), Collaborative decoding via Speculation (CoS) (Fu et al., 2025) theoretically guarantees that the inference speed is never slower than standard collaborative decoding. Since our UniR framework utilizes the summation of logits, we seamlessly apply the CoS approach to enhance inference speed. We employ the Llama model family and the reasoning module $\pi_r^{Math}$ trained on GSM8k dataset. To evaluate inference speed, we generate 100 tokens and measure the average token generation time across the entire GSM8k test benchmark.

## C ADDITIONAL EXPERIMENTS

### C.1 ANALYSIS OF THE REWARD DECOMPOSITION

To empirically validate our core assumption that the trajectory-level reward $r(x, y)$ can be expressed as the sum of log-probabilities from the reasoning module $\pi_r$ (Eq. 4), we evaluate our Llama-based model on the GSM8k test set. For each generated response, we compute $\sum_t \log \pi_r(y_t|x, y_{<t})$ using the reasoning module and group the responses according to whether the final answer is correct ($r = 1$) or incorrect ($r = 0$). To examine the effect of training, we compare outputs from the pre-trained module and the UniR-trained module.

As shown in Figure 5, correct responses consistently achieve higher summed log-probabilities than incorrect ones in both settings. This correlation provides empirical evidence that the reasoning module internalizes the external reward signal by assigning higher probability to trajectories that lead to correct outcomes. Moreover, while the pre-trained module already shows moderate separation between correct and incorrect responses, this separation becomes substantially more pronounced

after UniR training, indicating that our framework strengthens the module's ability to distinguish successful reasoning paths from unsuccessful ones.

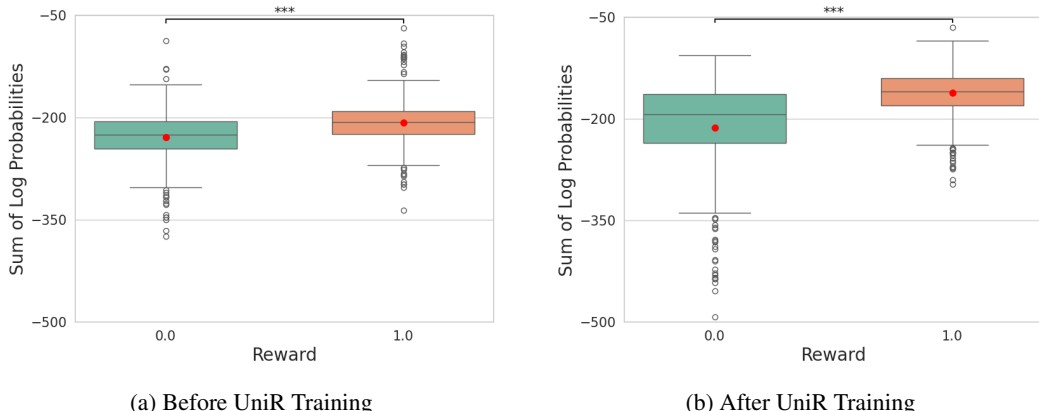

(a) Before UniR Training            (b) After UniR Training

Figure 5: Summed log-probability distributions from the reasoning module ($\pi_r$) on the GSM8k test set. (a) Before training, correct ($r = 1$) and incorrect ($r = 0$) responses exhibit moderate separation. (b) After UniR training, the separation becomes more pronounced, with correct responses receiving significantly higher probabilities. Red dots indicate means and asterisks denote statistical significance ($p < 0.001$).

## C.2    IMPACT OF REASONING MODEL SIZE

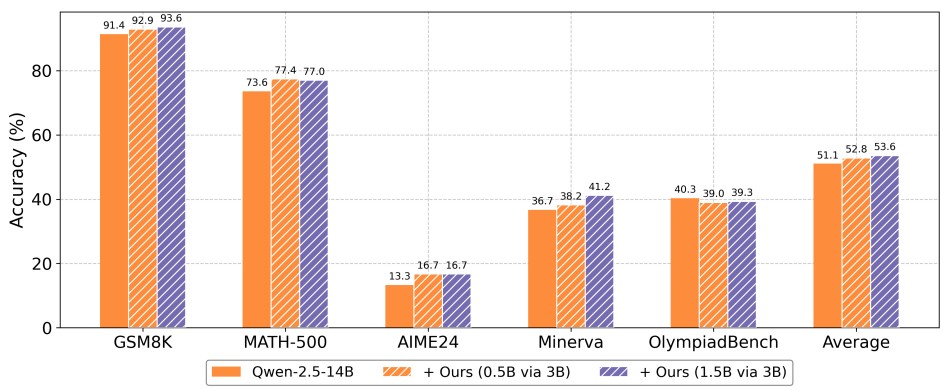

Figure 6: Transferability of the 0.5B $\pi_r$ and 1.5B $\pi_r$ reasoning modules when combined with a 14B frozen backbone model.

Table 9: Zero-shot Pass@1 performance on mathematical benchmarks comparing reasoning modules of different sizes (0.5B vs. 1.5B) trained with a 3B backbone.

| | Method | Trained Model | In-distribution | | Out-of-distribution | | | Avg. |
|---|---|---|---|---|---|---|---|---|
| | | | GSM8K | MATH-500 | AIME24 | Minerva | OlympiadBench | |
| Qwen2.5 3B | Backbone only | - | $74.4 \pm 1.0$ | $44.2 \pm 1.3$ | $6.3 \pm 3.7$ | $19.2 \pm 1.7$ | $24.6 \pm 0.9$ | 33.7 |
| | Backbone + 0.5B | - | $70.9 \pm 0.9$ | $43.1 \pm 1.2$ | $6.0 \pm 3.1$ | $16.5 \pm 1.9$ | $22.5 \pm 0.9$ | 31.8 |
| | **Ours** | 0.5B | $81.1 \pm 0.4$ | $62.2 \pm 1.4$ | $5.0 \pm 3.2$ | $24.7 \pm 2.0$ | $26.3 \pm 4.8$ | 39.9 |
| | **Ours** | 1.5B | $84.5 \pm 0.7$ | $66.8 \pm 1.4$ | $7.7 \pm 2.7$ | $26.3 \pm 2.0$ | $28.2 \pm 0.8$ | 42.7 |

We conducted an ablation study to investigate the impact of reasoning module size, training a Qwen2.5-0.5B reasoning module $\pi_r$ with a 3B backbone. As shown in Table 9, this smaller module achieved improved performance over the backbone alone, though it achieves lower performance

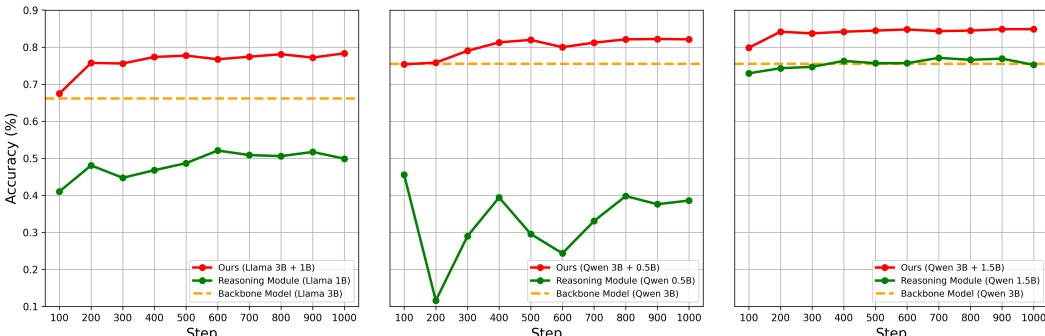

Figure 7: Performance comparison on the GSM8K dataset between standalone reasoning modules ($\pi_r$, green) and our UniR framework (red), evaluated against their respective backbone models (orange). (Left) $\pi_r$ module using Llama-3.2-1B with a Llama-3.2-3B backbone. (Middle) $\pi_r$ module using Qwen2.5-0.5B with a Qwen2.5-3B backbone. (Right) $\pi_r$ module using Qwen2.5-1.5B with a Qwen2.5-3B backbone.

than the larger 1.5B module. Figure 6 further illustrates the transferability of the 0.5B reasoning module $\pi_r$ to a 14B frozen backbone model $\pi_b$. The transferred 0.5B $\pi_r$ led to an improved average performance (52.8) compared to the baseline $\pi_r$ (51.1), showing that a module trained in a small-scale setting can generalize effectively to a larger model. Importantly, these results highlight that our main contribution is to demonstrate the ability to enhance the performance of larger models with comparatively small reasoning modules while prioritizing efficiency.

### C.3 ANALYSIS OF REASONING MODULE DURING TRAINING

We evaluated the performance of trained $\pi_r$ reasoning modules, as depicted in Figure 7. This figure tracks the GSM8k dataset accuracy throughout the training process of reasoning modules designed for two different backbone architectures. The results indicate that the reasoning modules, when used in isolation, exhibit suboptimal performance. However, when these same reasoning modules are combined with their respective backbone models within our proposed framework, the resulting performance surpasses that of the backbone model alone. This highlights the efficacy of our learned reasoning modules in providing valuable guidance when integrated with a larger language model, leading to enhanced reasoning capabilities in the combined system.

### C.4 GUIDANCE EFFECT OF $\pi_r$ ON BACKBONE INFERENCE.

Examining the reasoning capabilities of the $\pi_r$ and $\pi_b$ modules in isolation highlights both their individual limitations and the synergy achieved when combined. As shown in Figure 19, the standalone $\pi_r$ module fails to generate coherent and relevant reasoning for mathematical tasks. Likewise, the large backbone model alone struggles to reach correct conclusions on complex tasks. However, when the reasoning module is paired with a stronger backbone, it provides valuable guidance to the backbone's inference process, resulting in significantly improved reasoning performance.

### C.5 COMPUTATIONAL EFFICIENCY IN TRAINING

**VRAM Usage.** Due to its structural decomposability, our framework substantially reduces VRAM usage by avoiding gradient updates to the large backbone model. While LoRA also reduces memory usage by restricting updates to a small subset of parameters, it still incurs substantial activation memory costs since the full computation graph must be stored for backpropagation (Zhang et al., 2023). In contrast, our approach updates only the lightweight reasoning module, drastically minimizing activation memory requirements and enabling significantly larger batch sizes or sequence lengths under the same hardware constraints.

We examine peak memory usage as a function of the number of tokens processed per step. Input length was fixed to 250 tokens (plus up to 5 output tokens), and batch size was gradually increased

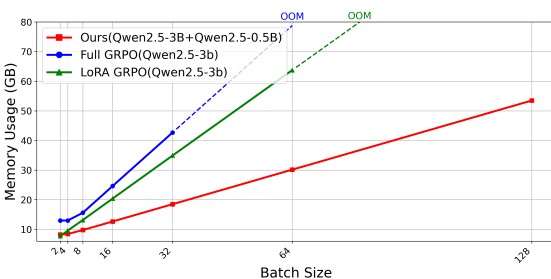

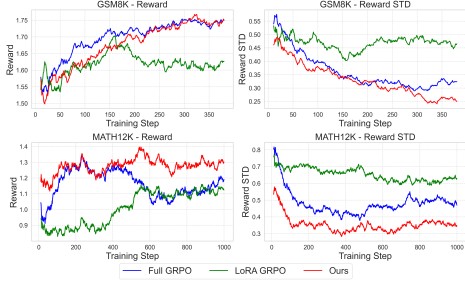

Figure 8: VRAM usage versus batch size under an 80GB constraint. Our method scales to batch size 128, while full fine-tuning and LoRA are limited, demonstrating memory efficiency for large batch.

Figure 9: Training dynamics. We visualize the mean (Left) and the standard deviation (Right) of reward on the GSM8k (Top) and Math-12k (Bottom) during training.

Table 10: Comparison of training throughput and inference FLOPs.

| Method | Training throughput (Sec. / step) ↓ | Inference FLOPs (GFLOPs) ↓ |
|---|---|---|
| Backbone | - | 6.17 |
| GRPO LoRA | 4.67 | 6.17 |
| GRPO Full | 7.87 | 6.17 |
| Ours | 6.29 | 7.16 |

under an 80GB VRAM limit. As shown in Figure 8, UniR supports batch sizes up to 128, whereas LoRA and full fine-tuning are limited to 64 and 32, respectively. Although sequence length was fixed in this experiment, activation memory generally scales with both batch size and sequence length, making UniR more flexible for large-scale training. Moreover, UniR remains compatible with efficiency techniques such as quantization, caching, and adapter-based methods (e.g., LoRA applied to the reasoning module itself).

**Training Throughput and Inference FLOPs.** To further demonstrate the efficiency of UniR, we report training throughput and inference FLOPs, computed as $FLOPs = num_{layers} \times (FLOPs_{Attn} + FLOPs_{MLP}) + 2 \times d_{model} \times vocab_{size}$, for UniR and GRPO variants under identical hardware using the Qwen model. As shown in Table 10, GRPO Full is slower than the other methods since it requires backpropagation through the entire 3B model in training. Although UniR is slightly slower than GRPO LoRA—mainly due to inference latency during reward computation—this bottleneck can be mitigated via Collaborative Speculation (CoS), as discussed in Sec. 5.4.

### C.6   REWARD STABILIZATION AND TRAINING DYNAMICS

**Reward variance.** A notable empirical finding during training is that UniR not only achieves higher rewards compared to GRPO but also exhibits a smaller reward variance, as shown in Figure 9. Since the advantage calculation in GRPO normalizes the unbiased advantage $r_i - \text{mean}(\{r_1, \ldots, r_G\})$ by its standard deviation (Eq. (7)), lower reward variance amplifies the magnitude of the effective advantage $A_i$. This means even small deviations from the mean reward translate into strong learning signals.

**Gradient norm.** We further analyze the gradient norm dynamics of UniR and GRPO Full over training steps on GSM8K and Math-500 using Qwen model. Specifically, we compute the average gradient norm every 100 training steps and report the maximum value within each 100-step window in parentheses in Table 11. These results demonstrate that UniR avoids the gradient explosions observed in GRPO Full, highlighting its training stability across both datasets.

These dynamics relate to the "Question-level difficulty bias" identified by (Liu et al., 2025b), where highly consistent rewards induce strong learning directions and thus lead to unstable policy updates. Indeed, GRPO often suffers from training instabilities when encountering policy collapse that can result in overfitting on *too easy* or *too hard* samples. By maintaining a fixed backbone model $\pi_b$,

Table 11: Gradient norm dynamics of GRPO Full and UniR on GSM8K and Math-500. We report average gradient norms every 100 training steps, with maximum values within each window shown in parentheses. Bold values indicate maximum gradient norms exceeding 5.

| Dataset | Method | 100 steps | 200 steps | 300 steps | 400 steps | 500 steps |
|---------|--------|-----------|-----------|-----------|-----------|-----------|
| GSM8K | GRPO Full | 1.44 (**27.6**) | 0.85 (2.74) | 0.74 (2.18) | 2.14 (**84.6**) | 93.9 (**3888.2**) |
| | Ours | 0.86 (1.50) | 0.74 (1.30) | 0.63 (1.19) | 0.70 (1.38) | 0.63 (1.19) |
| Math-500 | GRPO Full | 12.5 (**798.6**) | 21.1 (**1958.2**) | 1.18 (**42.5**) | 0.69 (3.77) | 0.77 (**7.11**) |
| | Ours | 0.69 (2.22) | 0.51 (1.32) | 0.48 (0.96) | 0.42 (1.31) | 0.47 (1.10) |

Table 12: Evaluation of using a small pre-trained reasoning model within the UniR framework. 'DeepSeek 1.5B' and 'Qwen 3B' represents DeepSeek-R1-Distill-Qwen-1.5B and Qwen2.5-3b-Instruct model, respectively.

| Backbone | Reasoning Module | Train with UniR | In-distribution | | Out-of-distribution | | | Avg. |
|----------|------------------|-----------------|-----------------|-----------|----------------------|---------|---------------|------|
| | | | GSM8K | MATH-500 | AIME24 | Minerva | OlympiadBench | |
| DeepSeek 1.5B | - | X | 71.7 | 51.6 | 0.0 | 17.7 | 23.3 | 32.8 |
| Qwen2.5 3B | DeepSeek 1.5B | X | 76.4 | 69.6 | 10.0 | 27.6 | 35.0 | 43.7 |
| Qwen2.5 3B | DeepSeek 1.5B | O | 83.8 | 73.4 | 10.0 | 30.1 | 37.3 | 46.9 |

UniR avoids such collapses and produces consistent responses with lower reward variance. This structural stability allows UniR to turn strong learning signals into constructive updates, rather than unstable divergence.

### C.7 DETAILED RESULTS OF USING A PRE-TRAINED MODEL AS A REASONING MODULE

This section presents the full results for using a pre-trained reasoning model as guidance. Table 12 breaks down the performance for each benchmark, showing that this approach improves scores even without UniR fine-tuning, with further gains achieved after optimization.

## D DECLARATION OF LLM USAGE

Our core methodology involves utilizing pre-trained LLMs in two key roles: (1) as a frozen backbone/reference model ($\pi_b$) whose generation process is guided, and (2) potentially as the architecture for the smaller, trainable reasoning policy ($\pi_r$) itself. The interaction between these LLM components, specifically the combination of their logits to form a guided policy, is a central and original part of our proposed method. We also use LLMs as evaluation methods, as described in the paper. In addition, LLMs were used during the writing stage only for language polishing and expression refinement. They were not used for research ideation, methodological design, or initial draft writing. All conceptual contributions and scientific content are solely the responsibility of the authors.

## E PROMPT TEMPLATE

For all models in the Llama family, we modified the default chat template by removing the knowledge cutoff and the generation time, as these fields vary depending on the time of execution and thus hinder reproducibility, and are also irrelevant to the task. For models in the Qwen family, however, we retained the default chat template as provided.

### E.1 MATH REASONING TASK.

The prompt template for mathematical problems follows the style of DeepSeek-R1 (DeepSeek-AI, 2025; Liu et al., 2025b). Our system prompts were designed to encourage the model to articulate its reasoning process before providing a final answer, illustrated in Figure 10, 11. Specifically, for the Math-12k dataset using Qwen model in Figure 11, we instructed the model to output its answer

within a boxed LaTeX environment. This format was adopted to ensure that various valid answer formats were correctly recognized and evaluated.

## E.2 TRANSLATION TASK.

For translation, inspired by the findings of MT-R1-Zero (Feng et al., 2025), which show that training with and without explicit reasoning yields comparable performance, we use direct translation prompts without explicit reasoning instructions detailed in Figure 12.

## E.3 COMPOSITIONAL TASK.

We instructed the model to solve math problems written in German and to perform reasoning and solving in English. The reasoning process was placed within <think> tags, and the final answer was enclosed within <answer> tags, as illustrated in Figure 13. Using the prompt shown in Figure 14, we evaluated the responses with GPT-4.1-nano-2025-04-14 on two dimensions: translation quality and math accuracy. Translation quality measures how well the response was translated into English, while math accuracy assesses accuracy using ground truth for a given problem.

While rule-based evaluation is often used for math accuracy due to the structured format of answers, we chose to use an LLM-based evaluator. This is because the math module is explicitly trained to follow a specific output format, whereas the translation module is not. Therefore, using a rule-based approach would unfairly favor the math side. To ensure a fair comparison between translation and problem-solving accuracy, we opt for GPT-4.1-nano as an impartial judge capable of understanding semantic correctness beyond strict formatting.

## E.4 MEDICAL TASK.

For our medical tasks shown in Figure 15, we designed the system prompt to explicitly instruct the model to produce a single numeric label, which must be enclosed within <answer> and </answer> tags. The user prompt provides the specific context for each prediction task. This clear explanation helps the model understand the task and the correct output format, reducing the chance of an incorrect or uninterpretable response.

---

**The prompt for GSM8K**

**System Prompt:** A conversation between User and Assistant. The user asks a question, and the Assistant solves it. The assistant first thinks about the reasoning process and answer are enclosed within <think> </think> and <answer> </answer> tags, respectively. Your response should be in the following format: <think> \n Your reasoning here \n </think> \n<answer> \n answer here \n </answer>. The reasoning process Note that respond by English, NOT use other languages.

**User Prompt:** {*question*}

---

Figure 10: For GSM8K, the prompt specifies a reasoning-then-answer format, where the model is instructed to articulate its thought process within <think> tags, followed by the final answer enclosed in <answer> tags. *question* denotes the question of the benchmark.

# F QUALITATIVE EXAMPLES

## F.1 EXAMPLE OF MATH TASK

Based on Table 1, fine-tuning with GRPO demonstrates improved performance over the base model; however, it also exhibits a tendency to generate longer outputs, which is alongside the results observed in (Yu et al., 2025; Team et al., 2025; Arora & Zanette, 2025). In contrast, our proposed method not only surpasses the performance of both the base model and the GRPO-tuned model but also achieves this with a lower number of tokens to reach the final answer. Figure 16 contains detailed examples that clearly show that our model is able to reach the correct answer in fewer steps.

---

**The prompt for Math-12K**

**- Llama**

**System Prompt:** A conversation between User and Assistant. The user asks a question, and the Assistant solves it. The assistant first thinks about the reasoning process and answer are enclosed within <think> </think> and <answer> </answer> tags, respectively. Your response should be in the following format: <think> \n Your reasoning here \n </think> \n <answer> \n answer here \n </answer>.

**User Prompt:** {*question*}

**- Qwen**

**System Prompt:** A conversation between User and Assistant. The user asks a question, and the Assistant solves it. The assistant first thinks about the reasoning process and answer are enclosed within <think> </think> and <answer> </answer> tags, respectively. Your response should be in the following format: <think> \n Your reasoning here \n </think> \n <answer> \n \boxed{{your answer here }} \n </answer>.

**User Prompt:** {*question*}

---

Figure 11: In the Math-12K prompt format used with the Llama and Qwen models, answers follow a reasoning-then-answer structure using <think> and <answer> tags. With Llama, we observed frequent failures to generate valid \boxed{} expressions, so we chose to omit this syntax. In contrast, when using Qwen, we explicitly enclose the final answer within \boxed{}, prompting the model to express its answer in a boxed format.

---

**The prompt for Translation**

**- Translation DE to EN**

**System Prompt:** You are a helpful assistant that translates German to English.

**User Prompt:** Translate the following German sentence into English. Respond ONLY with the translation. German: {*source*}

**- Translation EN to DE**

**System Prompt:** You are a helpful assistant that translates English to German.

**User Prompt:** Translate the following English sentence into German. Respond ONLY with the translation. English: {*source*}

---

Figure 12: Prompt used for Translation German-to-English and English-to-German. *source* indicates the source sentences to be translated.

## F.2 EXAMPLE OF TRANSLATION TASK

As shown in Figure 17, our method generates more fluent and faithful translations in both directions. Compared to the base model and GRPO, our DE→EN translation correctly uses the natural phrase "What about...", aligning with the tone of the original. On the EN→DE side, only our method preserves the correct compound adjective "künstlich intelligenten", which is grammatically accurate and semantically precise. In contrast, baseline and GRPO variants either use awkward phrasing ("artificial intelligent system") or deviate from standard German compound structures. This demonstrates the effectiveness of our method in producing higher-quality, context-aware translations.

---

**The prompt for German math questions into English**

**System Prompt:** You are a helpful assistant that translates German math questions into English and solves them. The user asks a math question in German. You first think about the reasoning process and answer. Both should be in English. Your response should be enclosed within <think> </think> and <answer> </answer> tags, respectively. Your response should be in the following format: <think> \n Your reasoning here \n </think> \n <answer> \n answer here \n </answer>.

**User Prompt:** German Question: {*question*}

---

Figure 13: System prompt used for German to English Math task: specifies the reasoning-then-answer format with <think> and <answer> tags. *question* indicates the math question in German.

---

**The prompt for scoring German math questions into English**

**- Evaluating Translation Quality**

**System Prompt:** You are a linguistic evaluator. Your job is to assess how well the model's response is in English and whether it avoids using German. Score only the language usage, not the math correctness.

**User Prompt:** Here is a model's response to a German math question. Please evaluate how well it used English instead of German. Assign a score between 1 and 5:
- 1: Mostly or entirely in German
- 3: Mixed, partly German, partly English
- 5: Fully in English, fluent and natural
Response: {*response*} \n End your response with only the score after '####'. Example: #### 4

**- Evaluating Accuracy**

**System Prompt:** You are a correctness evaluator. You will assess the factual accuracy of a math answer. You will return: \n AnswerCorrect: <True or False>

**User Prompt:** Here is a model's response to a German math question.
Ground Truth Answer: {*gt_answer*} \n {*response*} \n Evaluate: \n - AnswerCorrect: <True or False>

---

Figure 14: Prompt used for scoring German-to-English Math task: Translation quality and Math Accuracy. *response* and *gt_answer* represent the model output and the ground truth of each math question, respectively.

## F.3 EXAMPLE OF COMBINING SPECIALIZED MODULES

Figure 18 presents an example of the response to a German math-solving in English task. We compare the results from the UniR framework using the math-specific reasoning module ($\pi_r^{Math}$), the DE→EN translation-specific reasoning module ($\pi_r^{Trans}$), and their combination. This example demonstrates that using only the math module fails to generate a response in English, while using only the translation module results in an incorrect solution. In contrast, incorporating both modules achieves both high translation quality and mathematical accuracy.

## F.4 EXAMPLE OF $\pi_r$ REASONING PERFORMANCE

Figure 19 illustrates a specific instance where the backbone model and the reasoning module fail to adequately comprehend the problem's context. In contrast, our combined approach demonstrates a correct and coherent reasoning process for the same problem, highlighting the synergistic benefit of integrating the specialized reasoning module with the backbone model for improved contextual understanding and accurate inference.

---

**The prompt for Medical Task**

**System Prompt:** A conversation between User and Assistant. The user asks a question, and the Assistant solves it. The assistant first thinks about the reasoning process and answer are enclosed within <think> </think> and <answer> </answer> tags, respectively. Your response should be in the following format: <think> \n Your reasoning here \n </think> \n <answer> \n {*single number answer*} \n </answer>.

**- Readmission Prediction**

**User Prompt:** Based on the patient's conditions, procedures, and medications, is the patient likely to be readmitted to the hospital within 15 days of discharge? (Answer with 1 for yes and 0 for no) \n {*patient information*}

**- Length of Stay Prediction**

**User Prompt:** Based on the patient's conditions, procedures, and medications, predict the patient's ICU length of stay by categorizing it into one of the following 10 classes:\n0: ICU stay shorter than 1 day\n1: ICU stay lasting 1 to 7 days\n2: ICU stay longer than 7 days but less than or equal to 14 days\n3: ICU stay longer than 14 days\nReturn a single number (0–3) indicating the predicted length of stay category\n {*patient information*}

**- Mortality Prediction**

**User Prompt:** Based on the patient's conditions, procedures, and medications, predict whether the patient is likely to die during a subsequent hospital visit. (Answer with 1 for yes and 0 for no) \n {*patient information*}

---

Figure 15: The model was instructed to output a single numeric label within <answer> tags. Every user prompt included a clear explanation of the numeric labels.

## F.5 EXAMPLE OF FAILURE CASES

The primary cause of failure was the misinterpretation of semantic context. As illustrated in Figure 22, the first case shows the model overlooking the constraint "each kid," resulting in an incorrect distribution of the remaining marshmallows. In the second case, the model misinterprets the phrase "3-piece place setting." UniR incorrectly treats "3-piece" as meaning three of each item, whereas the intended meaning is that one set consists of three distinct items. These cases reflect the fundamental challenge of accurately translating semantic nuances into arithmetic operations.

**Prompt:** Hannah's city is having a big display of fireworks for the 4th of July. They're going to set off 15 boxes of 20 fireworks each. Hannah's house is at the right angle to see 40% of the city's fireworks. Hannah will also set off 3 boxes of 5 fireworks each in her backyard. How many fireworks will Hannah see in total?

**Backbone Model:**
<think> First, calculate the total number of fireworks in the display. There are 15 boxes of 20 fireworks each, so the total number of fireworks is 15 * 20. Then, calculate the number of fireworks Hannah can see in the display, which is 40% of the total number of fireworks. Next, calculate the number of fireworks in the 3 boxes she sets off in her backyard, which is 3 * 5. Finally, add the number of fireworks she can see in the display to the number of fireworks she sets off in her backyard to get the total number of fireworks Hannah sees. </think> <answer> The total number of fireworks in the display is 15 * 20 = 300. Hannah can see 40% of these, which is 0.40 * 300 = 120 fireworks. The number of fireworks in the 3 boxes she sets off in her backyard is 3 * 5 = 15. Therefore, the total number of fireworks Hannah sees is 120 + 15 = 135. </answer>

**GRPO:**
<think> First, let's calculate the total number of fireworks in the display. There are 15 boxes of 20 fireworks each, so the total number of fireworks is 15 * 20. Then, we need to find out how many of these fireworks Hannah can see, which is 40% of the total. After that, we'll add the 3 boxes of 5 fireworks each that Hannah sets off in her backyard. The total number of fireworks Hannah sees will be the sum of the fireworks she can see in the display and the fireworks she sets off in her backyard. </think> <answer> The total number of fireworks in the display is 15 * 20 = 300. Hannah can see 40% of these, which is 0.40 * 300 = 120 fireworks. She also sets off 3 boxes of 5 fireworks each, which is 3 * 5 = 15 fireworks. Therefore, the total number of fireworks Hannah sees is 120 + 15 = 135. </answer>

**Ours:**
<think> First, let's calculate the total number of fireworks in the display. There are 15 boxes of 20 fireworks each, so the total number of fireworks is 15 * 20 = 300. Hannah's house is at the right angle to see 40% of the city's fireworks, so the number of fireworks she will see is 300 * 0.4 = 120. In her backyard, she will set off 3 boxes of 5 fireworks each, which totals 3 * 5 = 15 fireworks. Therefore, the total number of fireworks Hannah will see is 120 + 15 = 135. </think> <answer> 135 </answer>

Figure 16: Chain-of-thought comparison on a GSM8k example: while the Backbone only and GRPO models both arrive at the correct answer with lengthy explanations, our approach produces the same result with a much more concise reasoning process.

**Source Prompt:**
EN : What about the artificially intelligent system in the movie "Her," Samantha?
DE : Was ist mit dem künstlichen intelligenten System im Film "Her", Samantha?

**Backbone Model:**
DE → EN: What's with the artificial intelligent system in the movie "Her", Samantha?
EN → DE: Was ist mit dem künstlichen Intelligenzsystem im Film "Her", Samantha?

**GRPO:**
DE → EN: What's with the artificial intelligence system in the movie "Her"?
EN → DE: Was ist mit dem künstlichen Intelligenzsystem in dem Film "Her", Samantha?

**Ours:**
DE → EN: What about the artificial intelligence system in the movie "Her," Samantha?
EN → DE: Was ist mit dem künstlich intelligenten System im Film "Her", Samantha?

Figure 17: Example of bidirectional translation (German (DE) and English (EN)). Our approach produces more fluent and grammatically correct outputs in both directions, preserving natural phrasing and accurate compound structures.

---

**Prompt:**

Billy schält und schneidet Kartoffeln als Vorbereitung für die Zubereitung des Kartoffelsalats für seine große Grillparty anlässlich eines Familientreffens. Es ist ein großes Ereignis, also muss er 60 Kartoffeln schaffen. Die Kartoffeln haben ungefähr die gleiche Größe, also braucht es ungefähr gleich viel Zeit, um jede Einzelne zu schälen und zu schneiden. Billy braucht ungefähr eineinhalb Minuten, um eine Kartoffel zu schälen, aber nur ungefähr 5 Sekunden, um sie klein zu schneiden. Wie lange braucht Billy, um die Kartoffelvorbereitung abzuschließen?

---

**Backbone Model + $\pi_r^{Math}$:**

<think> Zunächst müssen wir die Gesamtzeit berechnen, die Billy benötigt, um die Kartoffeln zu schälen und zu schneiden. Da es 60 Kartoffeln gibt und jede Kartoffel ungefähr die gleiche Größe hat, können wir die Zeit pro Kartoffel berechnen. Die Zeit, die Billy benötigt, um eine Kartoffel zu schälen, beträgt ungefähr 1,5 Minuten, also 1,5 * 60 = 90 Sekunden. Die Zeit, die Billy benötigt, um eine Kartoffel zu schneiden, beträgt ungefähr 5 Sekunden. Um die Gesamtzeit zu berechnen, müssen wir die Zeit pro Kartoffel für die Schälen und Schneiden addieren. Die Gesamtzeit beträgt also 90 + 5 = 95 Sekunden. Da es 60 Kartoffeln gibt, müssen wir die Gesamtzeit mit 60 multiplizieren, um die Gesamtzeit für die Vorbereitung aller Kartoffeln zu erhalten. Die Gesamtzeit beträgt also 60 * 95 = 5700 Sekunden. Um die Zeit in Minuten umzurechnen, können wir die Sekunden durch 60 teilen. Die Gesamtzeit beträgt also 5700 / 60 = 95 Minuten. </think> <answer> 95 Minuten </answer> </answer>

---

**Backbone Model + $\pi_r^{Trans}$:**

<think> To solve this problem, we need to calculate the total time it takes Billy to prepare the carrots for his party. Since he needs to prepare 60 carrots, and the time it takes to prepare one carrot is 1.5 minutes, we can multiply the number of carrots by the time it takes to prepare one carrot. However, we also need to consider the time it takes to cut the carrots into smaller pieces, which is 5 seconds per carrot. So, we need to add the time it takes to cut the carrots to the total time. First, let's calculate the time it takes to prepare one carrot: 1.5 minutes = 90 seconds. Now, let's calculate the time it takes to cut one carrot: 5 seconds. Since Billy needs to cut 60 carrots, we can multiply the time it takes to cut one carrot by 60: 5 seconds * 60 = 300 seconds. Now, let's add the time it takes to prepare one carrot to the time it takes to cut one carrot: 90 seconds + 300 seconds = 390 seconds. Finally, let's convert the total time from seconds to minutes: 390 seconds / 60 = 6.5 minutes. So, Billy needs 6.5 minutes to prepare the carrots for his party. </think> <answer> Billy needs 6.5 minutes to prepare the carrots for his party.

---

**Backbone Model + $0.5\pi_r^{Math}$ + $0.5\pi_r^{Trans}$:**

<think> First, let's calculate the total time it takes to peel and chop one potato. It takes Billy 1.5 minutes to peel a potato, and 5 seconds to chop it. To convert the time to seconds, we multiply 1.5 minutes by 60 seconds per minute, which equals 90 seconds. Now, we add the time it takes to peel and chop one potato, which is 90 seconds + 5 seconds = 95 seconds. Since Billy needs to peel and chop 60 potatoes, we multiply the time it takes to peel and chop one potato by 60, which equals 95 seconds * 60 = 5700 seconds. To convert this to minutes, we divide by 60, which equals 5700 seconds / 60 = 95 minutes. Therefore, Billy needs 95 minutes to peel and chop all 60 potatoes. </think> <answer> 95 minutes </answer>

---

Figure 18: We illustrate outputs for a German math problem solved in English. (Top) Guidance with only the math module solves correctly but responds in German; (Middle) while using only a translation module ensures English output but fails at math. (Bottom) Their combination achieves both target language and correct solution.

**Prompt:**
Tanya is teaching at school. She earns $15 for every hour and an additional $5 per day if she teaches more than 3 classes. On Monday she teaches 4 classes for 5 hours, and on Wednesday 2 classes for 2 hours. How much did Tanya earn for these two days of teaching?

**Backbone Model $\pi_b$:**
`<think>` First, let's calculate the earnings for Monday. Tanya teaches 4 classes for 5 hours each. Her base pay for teaching is $15 per hour. So, her earnings from teaching alone on Monday would be 4 classes * 5 hours/class * $15/hour....`</think><answer>` For Monday, Tanya's earnings from teaching are 4 * 5 * 15 = $300. Since she teaches more than 3 classes, she earns an additional $5, so her total for Monday is $300 + $5 = $305. For Wednesday, Tanya's earnings from teaching are 2 * 2 * 15 = $60....The answer is $365.`</answer>`

**Reasoning Module $\pi_r$:**
`<think> <answer>  <ul>` `<li>`On Monday, she earns $15/hour * 4 classes * 5 hours = $300`</li> <li>` On Monday, she earns $15/hour * 2 classes * 2 hours = $60`</li> <li>` On Wednesday, she earns $15/hour * 2 classes * 2 hours = $60`</li> <li>` On Wednesday, she earns $15/hour * 2 classes * 2 hours = $60`</li>` `<li>`On Monday, she earns $15/hour * 4 classes * 5 hours = $300`</li> <li>` On Monday, she earns $15/hour * 2 classes * 2 hours = $60 `</li> <li>` On Wednesday, she earns $15/hour * 2 classes * ...

**Ours $\pi_b + \pi_r$:**
`<think>` First, let's calculate the earnings for Monday. Tanya teaches 4 classes for 5 hours each, so she earns $15 per hour for 5 hours, which is $15 * 5 = $75. Additionally, since she teaches more than 3 classes, she earns an extra $5 per day, which is $5. So, her total earnings for Monday are $75 + $5 = $80. Next, ... Now, we add the earnings from both days to get the total earnings for the two days: $80 (Monday) + $30 (Wednesday) = $110.`</think><answer>` $110 `</answer>`

Figure 19: **Reasoning Performance of the $\pi_r$ Module.** (Left) The backbone model $\pi_b$ and (Middle) the standalone reasoning module $\pi_r$ produce incorrect, repetitive and logically flawed reasoning. (Right) When combined, it generates coherent reasoning and arrive at the correct solution, showing the effectiveness of the modular guidance.

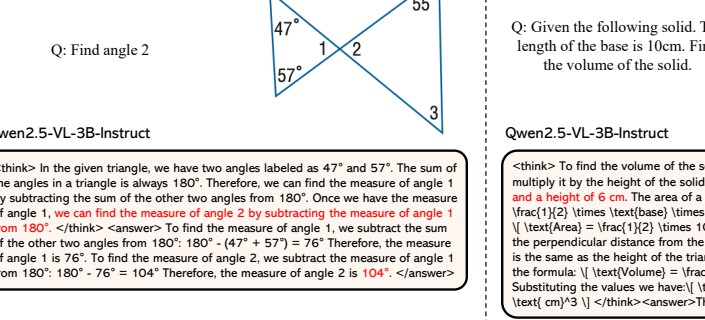

Q: Find angle 2

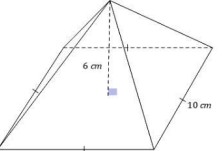

Q: Given the following solid. The length of the base is 10cm. Find the volume of the solid.

Qwen2.5-VL-3B-Instruct

`<think>` In the given triangle, we have two angles labeled as 47° and 57°. The sum of the angles in a triangle is always 180°. Therefore, we can find the measure of angle 1 by subtracting the sum of the other two angles from 180°. Once we have the measure of angle 1, we can find the measure of angle 2 by subtracting the measure of angle 1 from 180°. `</think> <answer>` To find the measure of angle 1, we subtract the sum of the other two angles from 180°: 180° - (47° + 57°) = 76° Therefore, the measure of angle 1 is 76°. To find the measure of angle 2, we subtract the measure of angle 1 from 180°: 180° - 76° = 104° Therefore, the measure of angle 2 is 104°. `</answer>`

Qwen2.5-VL-3B-Instruct

`<think>` To find the volume of the solid, we need to determine the area of the base and then multiply it by the height of the solid. The base of the solid is a triangle with a base of 10 cm and a height of 6 cm. The area of a triangle is given by the formula: \[ \text{Area} = \frac{1}{2} \times \text{base} \times \text{height} \] Substituting the given values: \[ \text{Area} = \frac{1}{2} \times 10 \times 6 = 30 \text{ cm}^2 \] The height of the solid is the perpendicular distance from the base to the apex. Since the solid is a pyramid, the height is the same as the height of the triangle, which is 6 cm. The volume of a pyramid is given by the formula: \[ \text{Volume} = \frac{1}{3} \times \text{Base Area} \times \text{Height} \] Substituting the values we have:\[ \text{Volume} = \frac{1}{3} \times 30 \times 6 = 60 \text{ cm}^3 \] `</think><answer>`The volume of the solid is 60 cm³. `</answer>`

UniR ( + Qwen2.5-0.5B-Instruct )

`<think>` The given figure is a triangle with angles labeled as follows: $\angle 1 = 47^\circ$, $\angle 2 = 55^\circ$, and $\angle 3 = 57^\circ$. The sum of the angles in a triangle is $180^\circ$. Therefore, we can find $\angle 2$ by subtracting the sum of the other two angles from $180^\circ$. \[ \angle 2 = 180^\circ - (\angle 1 + \angle 3) = 180^\circ - (47^\circ + 57^\circ) = 180^\circ - 104^\circ = 76^\circ \] `</think>` `<answer>` 76^\circ `</answer>`

UniR ( + Qwen2.5-0.5B-Instruct )

`<think>`To find the volume of the solid, we need to determine the area of the base and the height of the solid. The base is a rectangle with a length of 10 cm and a width of 10 cm, so the area of the base is \(10 \times 10 = 100 \, \text{cm}^2\). The height of the solid is the perpendicular distance from the base to the apex, which is given as 6 cm. The volume \(V\) of a pyramid is given by the formula \(V = \frac{1}{3} \times \text{Base Area} \times \text{Height}\). Substituting the values, we get \(V = \frac{1}{3} \times 100 \times 6 = 200 \, \text{cm}^3\). `</think>` `<answer>` 200 `</answer>`

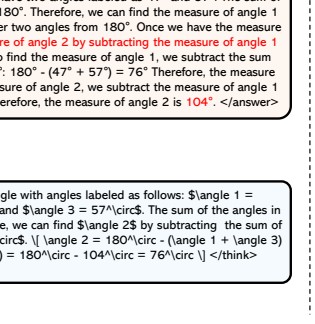

(a) Geometry3k

(b) MathVerse

Figure 20: Illustrative examples of responses from the baseline VLM and our UniR-extended framework on tasks from the (Left) Geometry3k and (Right) MathVerse benchmarks.

---

**Patient Information:**

Patient ID: ****

Visit 0:

Conditions:

1. other complications of birth; puerperium affecting management of mother
2. early or threatened labor
3. other pregnancy and delivery including normal

...

Procedures:

1. cesarean section

Medications:

1. vitamin b 12 and folic acid
2. antiinfectives and antiseptics, excl. combinations with corticosteroids
3. lipid modifying agents, plain

...

---

**Backbone Model:**

<think> The patient has a history of complications related to pregnancy and delivery, which could indicate a high risk of readmission. The patient also has a history of mycoses, which could be a sign of an underlying health issue that might require further medical attention. The patient has also undergone a cesarean section, which is a significant procedure that could lead to complications. The patient is on a variety of medications, including corticosteroids, antacids, and analgesics, which could indicate ongoing health issues. However, the list of medications does not include any specific medications that are typically used for readmission prevention. </think> <answer> 0 </answer>

---

**Backbone Model + $\pi_r^{Read}$:**

<think> To determine if the patient is likely to be readmitted to the hospital within 15 days of discharge, I need to consider several factors including the patient's current health status, the procedures and medications they are undergoing, and the conditions they are currently experiencing. The patient's conditions, such as complications of pregnancy, early or threatened labor, and other complications of birth, suggest that they are in a high-risk category for readmission. The procedures, such as cesarean section, also indicate a high-risk situation. The medications, including antiinfectives and antiseptics, and lipid modifying agents, suggest that the patient is at risk for infections and other complications. The patient's current medications, such as vitamin B12 and folic acid, and iron antianemic preparations, suggest that they are at risk for anemia. The patient's current medications, such as corticosteroids for systemic use, and other analgesics and antipyretics, suggest that they are at risk for side effects and complications. The patient's current medications, such as antipruritics, antihistamines, and throat preparations, suggest that they are at risk for allergic reactions and other side effects. The patient's current medications, such as digestive enzymes, suggest that they are at risk for digestive issues. </think> <answer> 0 </answer>

---

**Backbone Model + $\pi_r^{Read}$ + $\pi_r^{LOS}$:**

<think> The patient is likely to be readmitted to the hospital within 15 days of discharge due to the presence of multiple complications and procedures, including cesarean section, which can lead to postoperative complications. The patient's conditions and medications, such as vitamin B12 and folic acid, antiinfectives, and lipid modifying agents, are also relevant to the patient's health status. The patient's history of early or threatened labor, mycoses, esophageal disorders, and headache, including migraine, also suggest a high risk of readmission. Additionally, the patient's use of corticosteroids for systemic use, plain, and other analgesics and antipyretics in atc, as well as the use of beta-lactam antibacterials, penicillins, and other therapeutic products, further increases the likelihood of readmission.</think> <answer> 1 </answer>

---

Figure 21: Example of Readmission Prediction. (Top) the output of the backbone model; (Middle) the backbone model with the Readmission module; (Bottom) and the backbone model with both the Readmission and Length of Stay modules. The model with both specialized modules successfully identifies risk factors and provides an accurate final prediction.

**Prompt:**
John and DeSean bought a bag of marshmallows to make s'mores together. The bag has 35 marshmallows. Each S'more uses one marshmallow. If John makes 9 S'mores, DeSean makes 9 S'mores, and they dropped 3 marshmallows on the ground, how many S'mores can each kid have with the marshmallows left in the bag?

**Ours:**
<think> First, let's determine the total number of marshmallows used by John and DeSean. Each S'more uses one marshmallow, so together they made 9 + 9 = 18 S'mores. They started with 35 marshmallows and dropped 3 on the ground, so they have 35 - 3 = 32 marshmallows left. Next, we need to find out how many marshmallows are left after they've made their S'mores. Since they used 18 marshmallows, the remaining marshmallows are 32 - 18 = 14. Now, we need to determine how many S'mores each can make with the remaining marshmallows. Since each S'more uses one marshmallow, each can make 14 S'mores with the remaining marshmallows. Therefore, John and DeSean can each make 14 S'mores with the marshmallows left in the bag. </think> <answer> 14 </answer>

**Prompt:**
Avery needs to buy a 3 piece place setting (dinner & salad plate and a bowl) for her holiday dinner. She's having 12 people over for dinner. If the dinner plates cost $6.00 each and bowls each cost $5.00 and the salad plates cost $4.00, how much will she spend on place settings?

**Ours:**
<think> To determine the total cost of the place settings, we need to calculate the cost for each type of plate and bowl and then sum them up.1. Dinner Plates: - Avery needs 3 dinner plates. - Each dinner plate costs $6.00. - Total cost for dinner plates = 3 * $6.00 = $18.00. 2. Salad Plates: - Avery needs 3 salad plates. - Each salad plate costs $4.00. - Total cost for salad plates = 3 * $4.00 = $12.00. 3. Bowls: - Avery needs 3 bowls. - Each bowl costs $5.00. - Total cost for bowls = 3 * $5.00 = $15.00. Now, we sum up the costs of all the place settings:- Total cost = Cost of dinner plates + Cost of salad plates + Cost of bowls- Total cost = $18.00 + $12.00 + $15.00 = $45.00.Therefore, Avery will spend $45.00 on place settings.</think> <answer>$45.00 </answer>

Figure 22: Examples of Failure Cases

