# OpenReview forum: "Universal Reasoner: A Single, Composable Plug-and-Play Reasoner for Frozen LLMs"
_ICLR.cc/2026/Conference — Submitted to ICLR 2026_

### Official Review · Reviewer_8VRM · 2025-10-31

**Soundness:** 3
**Presentation:** 4
**Contribution:** 3
**Rating:** 6
**Confidence:** 3

**Summary:**

This paper proposes a method to decouple reasoning from the base model by introducing a standalone, pluggable reasoning module that can be integrated with any model size or architecture by adding its output logits to those of the base model. The paper shows that it is possible to compose even two reasoning modules to combine task-specific abilities, achieving improved performance over both the base model and the GRPO-trained model (in math and translation tasks).

**Strengths:**

1. A decoupled reasoning module that is model- and architecture-agnostic (assuming tokenizers can be aligned) with the base model, demonstrating that reasoning can be isolated and trained independently.

2. The reasoning module can be seamlessly integrated by adding its output logits to those of the base model, and it appears to enhance performance even in much larger base models.

3. A linear composition of reasoning modules is possible and effective, generalizing well even to unseen test sets (e.g., mortality prediction).

4. The paper is well presented and demonstrates strong understanding and integration of relevant theoretical concepts.

**Weaknesses:**

1. How are the coefficients determined when combining reasoning modules?

2. Is it possible to compose more than two modules, and if so, how does performance scale with additional compositions?

**Questions:**

1. Does the reasoning module require any SFT warm start? Prior studies have shown that directly starting with GRPO, especially on small models, often leads to suboptimal performance.

---

> ### Author Response · Authors · 2025-11-22
>
> **W1 & W2. Composition Scaling and Performance**
>
> In Table 1, 2 and Figure 2, 4, we fix the coefficient as $\alpha=1.0$. We will clarify this in the Table 7. In Figure 3, we scale $\alpha$ from 0 to 1.0 to analyze the combination ratio between the math and translation modules, and the performance changes according to $\alpha$ can be observed in Figure 3. For Table 3, $\alpha=1.0$ is used, as specified in Appendix B.3.1.
>
> We also experimented with composing three separately trained reasoning modules -- mortality, readmission, and length of stay. While this simple combination of UniR modules is unlikely to outperform the dedicated module for each individual task, it may enable the model to acquire broader cross-task knowledge compared to the baseline. The results are summarized below:
>
> | Task | Method | Reasoning Module | Acc. | F1 |
> | -- | -- | -- | -- | -- |
> | Readmission (R) | Backbone only | - | 39.46 | 25.87 |
> | | UniR | R | 52.91 | 26.77 |
> | | UniR | R + L + M | 35.24 | 25.35 |
> | Length of stay (L) | Backbone only | - | 9.64 | 5.92 |
> | | UniR | L | 67.77 | 16.25 |
> | | UniR | R + L + M | 54.42 | 17.35 |
> | Mortality (M) | Backbone only | - | 39.76 | 28.27 |
> | | UniR | M | 81.12  | 46.76  |
> | | UniR | R + L + M | 59.44 | 31.41 |
>
> This multi-task architecture  achieves significant performance improvements over the backbone model alone for the tasks, such as length of stay (9.64 → 54.42) and mortality (39.76 → 59.44), except for the readmission task, which shows the comparable results. In a readmission task, this outcome is hypothesized to stem from the fact that predicting readmission task requires distinct set of knowledge compared to  length of stay and mortality, which can have complementary effect.
>
> **Q1. Question about SFT warm start**
>
> No, the reasoning modules ($\pi_r$) used in our experiments did not apply any task-specific SFT warm start. They were initialized from their pre-trained checkpoints.
>
> However, we agree with the reviewer's intuition that a stronger initialization is beneficial. We explored this in Sec 5.4 (Table 5), where we used a pre-trained reasoning model (DeepSeek-R1-Distill-Qwen-1.5B) as the $\pi_r$ module. As the reviewer anticipated, this stronger prior improved performance, and our UniR training further enhanced it. This confirms that while UniR does not require SFT, its performance can be further boosted by initializing $\pi_r$ with a stronger task-aligned model.

---

### Official Review · Reviewer_PpAD · 2025-11-01

**Soundness:** 3
**Presentation:** 3
**Contribution:** 2
**Rating:** 6
**Confidence:** 3

**Summary:**

This paper studies LLM reasoning tasks and proposes UniR, a standalone reasoning module that reduces training costs while achieving generalization across models. UniR is composable and trained via GRPO. It provides token-level logit guidance to the backbone LLM. UniR demonstrates better performance than GRPO LoRA and GRPO Full with lower training costs. Additionally, UniR trained on a smaller backbone can improve task performance when combined with a larger backbone.

**Strengths:**

This paper has a clear motivation: reduce computational costs in post-training and improve the generalization of learned reasoning abilities.

The content is high quality and the writing is clear and easy to understand.

The experiments are extensive, covering various benchmarks, tasks, model sizes, and series. The proposed UniR shows better performance compared with baseline methods and improves performance on larger backbones that were not used to train UniR.

When combined, different UniR modules show adjustable performance improvements on math and translation tasks.

**Weaknesses:**

One weakness is the lack of detailed discussion about time and resource costs. The paper should highlight how UniR differs from baseline methods to demonstrate its computational advantages.

Another point: the paper needs more analysis of the design choices for the standalone reasoning module. The authors use a smaller LLM as the reasoning module, which is a good initial step. However, have they experimented with using parts of a model instead of a full LLM? Could the reasoning module be further reduced in size?

Additionally, more analysis comparing outputs from baseline GRPO versus UniR would better illustrate UniR's influence. For example, what helpful adjustments does token-level logit modification provide compared to optimizing the model using only the final reward signal?

**Questions:**

Regarding the first contribution (Line 75–76): circumventing the need for expensive preference dataset creation is not a major contribution of this work. The tasks studied here do not require such preference dataset construction.

In Figure 3, what is the baseline performance—that is, only the base model without the reasoning module? Showing the baseline performance would help readers better understand this figure.

---

> ### Author Response · Authors · 2025-11-22
>
> **W1. Computational Cost**
>
> We thank the reviewer for this important point. We have provided detailed analyses of computational costs in Sec.5.4 (Table 6) and Appendix C.5 (Figure 8, Table 10), which we summarize here.
>
> As shown in Figure 8, our primary computational advantage during training is VRAM efficiency. While GRPO-Full must backpropagate through the entire large backbone model, and GRPO-LoRA must store the full computation graph for the frozen backbone, UniR only requires gradients and activation memory for the lightweight reasoning module. Due to its efficiency, UniR (6.29 sec/step) is faster than GRPO Full (7.87 sec/step) in terms of throughput (Table 10).
>
> In addition, we integrated UniR with a collaborative speculative decoding method (CoS) which brought inference speed (0.0279 sec/token) close to that of the backbone model (0.0262 sec/token) (Table 6).
>
> **W2. Design Choice of Reasoning Module**
>
> We would like to highlight that the reasoning module can be reduced in size (Qwen 1.5B → 0.5B), achieving performance improvements (Table 9) while enabling transferability (Figure 6) in appendix.
>
> To investigate if a full small LLM is necessary, we conducted an experiment using a minimal part as the reasoning module, as you suggested. We duplicated the LM head of the frozen Qwen 3B backbone (~300M parameters) and trained only this new head using our GRPO objective, initializing it from the original LM head weights. The final logit was the sum of the original frozen head's logit and this new reasoning head’s logit.
>
> | Method | GSM8K | MATH500 | AIME | Minerva | Olympiad
> | -- | -- | -- | -- | -- | -- |
> |Baseline only 	| 74.4 | 44.2 | 6.3 | 19.2 | 24.6|
> |UniR (LM head) | 76.8 | 49.6 | 3.3 | 22.1 | 26.6|
> |UniR (1.5B) | 84.5 | 66.8 | 7.7 | 26.3 | 28.2 |
>
> As the results show, this reasoning LM head module improved the GSM8K (74.4 → 76.8) and MATH500 (44.2 → 49.6), confirming that a minimal module can learn and provide effective guidance. However, this performance is still lower than that achieved by our main UniR method (84.5 and 66.8).
>
> We attribute this to the limited expressive power of a single linear layer. Furthermore, this framework does not support transferring to other backbone LLMs with different hidden state dimensions. This experiment suggests that the multi-layer architecture of a small-but-full LLM is beneficial for capturing the more complex token-level reasoning guidance.
>
> **W3. Comparison of GRPO, Backbone Model and UniR**
>
> The key difference lies in the correctness of the reasoning process itself. As detailed in Appendix Figures 16 and 17, models like the backbone-only and GRPO frequently show errors within their reasoning steps. In contrast, UniR consistently demonstrates a more accurate reasoning process throughout.
>
> Furthermore, to quantify the effect of token-level adjustments, we compared the summation of the output log-probability distributions of the pre-trained module and the UniR-trained module (detailed in Appendix Figure 5). The analysis shows that after training with UniR, the distributional gap between correct and incorrect answers becomes wider. This change illustrates that UniR's token-level logit modification by trained reasoning module strengthens the model’s fundamental ability to distinguish between successful and unsuccessful reasoning paths.

---

> > ### Author Response · Authors · 2025-11-22
> >
> > **Q1. Question of Contribution**
> >
> > Thank you for clarifying our contribution; our primary tasks rely on verifiable rewards, not human preference datasets. We would like to highlight that our intended contribution was not to claim we invented an alternative to preference data. Rather, our contribution is the method of efficiently and directly using verifiable rewards within the GRPO framework to provide token-level guidance. This contrasts with other token-level guidance methods like GenARM, which are trained on preference datasets. Our framework is specifically tailored for domains where objective rewards are available.
> > We revise the phrasing in our introduction and contribution list to make this distinction clear.
> >
> > **Q2. Baseline Performance in Figure 3**
> >
> > We thank the reviewer’s suggestion for including the baseline performance which provides a crucial insight into how our UniR modules function. We add this data point to Figure 3 in the revised version.
> >
> > The performance of the Backbone-only model on this task (German math problems $\to$ English solution) is:
> >
> > | Method | Math Acc. | Trans. Quality |
> > | -- | -- | -- |
> > | Backbone-only | 20.4 % | 4.98 |
> > | $\alpha = 0$ | 57.6 % | 4.92 |
> >
> > The baseline model produces a high-quality English translation of the German question (4.98 quality score), but it fails to solve the mathematical problem itself (20.4% accuracy). At first, it seems contradictory that adding a Translation module ($\alpha=0$) dramatically improves Math Accuracy.
> >
> > This is the key insight of our logit-addition method. The $\pi^\text{Trans}_r$ module's token-level guidance does not just translate, rather it steers the frozen backbone's internal reasoning state into a fluent English-speaking mode. Once the backbone is controlled to think in coherent English, its own mathematical ability which is confused by the German input is improved and causes the Math Accuracy to jump from 20.4% to 57.60%. Since the model's output is no longer a literal translation of the question, the slight drop in Translation Quality (4.98 → 4.92) is also explained.

---

### Official Review · Reviewer_xWTC · 2025-11-03

**Soundness:** 3
**Presentation:** 3
**Contribution:** 4
**Rating:** 6
**Confidence:** 3

**Summary:**

The paper proposes a single "universal" model that converts practically any LLM into a reasoning-enabled LLM by adding logits at inference time. This enables reasoning behavior without the cost of fine-tuning and without the reduction in quality that reasoning-fine-tuning often produces.

**Strengths:**

This model seems very useful both as a research artifact (it seems to tell us that there isn't all that much to reasoning behavior; not many weights are actually needed to represent it "from scratch") as well as to users of AI tech. Enabling a "plug-and-play" ecosystem of foundation models and reasoning models could have a high impact on the community.

**Weaknesses:**

Some of the baseline numbers are suspicious. For example, this paper reports a "backbone only" performance on MATH of 28.7 for llama3.2 3B, but huggingface here (https://huggingface.co/meta-llama/Llama-3.2-3B-Instruct) reports a 48.0 accuracy and that's consistent with what ArtificialAnalysis reports (https://artificialanalysis.ai/evaluations/math-500).  I'm also having trouble finding the other baseline numbers in prior work or online (GSM8K seems to mostly be 8-shot in evaluations of llama3 as opposed to the 0-shot evaluated in the paper).

**Questions:**

Are there sources that can corroborate that the baseline numbers in Table 1 and Table 2 are correct (i.e. what we should expect for these models)?

Can you go into more detail about the architecture of the universal reasoner? The paper is vague on this point.

---

> ### Author Response · Authors · 2025-11-22
>
> **W1 & Q1. Clarifying and correcting backbone baselines**
>
> We thank the reviewer for carefully checking the baseline numbers. After re-examining our evaluation logs, we confirmed that the originally reported MATH baseline for Llama-3.2-3B (28.7 ± 9.9) contained an experimental error. Two of the ten seeds terminated prematurely due to an experimental issue, and their incomplete outputs were inadvertently included in the average. We apologize for this oversight and for the resulting confusion. After re-running the affected seeds to completion, an accurate baseline is 33.0 ± 2.3. Importantly, the correction does not change any of our main conclusions, and UniR still consistently outperforms both GRPO-LoRA and GRPO-Full under the same evaluation settings. We have updated Table 1 and the corresponding text to reflect corrected value, and we have carefully re-checked all other results in Tables 1 and 2 and did not identify further issues.
>
> Even with this correction, our backbone numbers remain lower than those reported by HuggingFace or ArtificialAnalysis. The remaining discrepancy is expected and arises from (i) different prompting and evaluation setup, (ii) the use of stochastic decoding in the main tables.
>
> - **Prompting setup**: We use a DeepSeek-R1–style zero shot chain-of-thought prompt (Appendix E.1), rather than the task-specific prompts used in public leaderboards. We intentionally reused a zero-shot CoT template across the math benchmarks to simplify our methodology, and it means that our baselines are not comparable to numbers reported under optimized, benchmark-specific prompts.
> - **Sampling temperature**: The main results (Table 1,2) report with temperature 0.6 averaged over 10 seeds to capture the statistical behavior of the models. As expected, high temperature reduces performance relative to greedy decoding (temperature 0). The tables below report the baseline and our method under greedy decoding. As shown, most scores are higher than those in the main tables. For example, under our prompt and greedy decoding, the Llama-3.2-3B baseline achieves additional 5 percent points (38.0%) on MATH.
>
> **Llama Math**
> | Method | GSM8K | MATH | AIME | Minerva | OlympiadBench | Avg. |
> | -- | -- | -- | -- | -- | -- | -- |
> | Backbone only | 66.1| 38.0 | 0.0 | 14.8 |  11.4 |  26.1|
> | Ours     |78.3 | 49.2 | 16.7 | 18.4 | 17.3  | 36.0 |
>
> **Qwen Math**
> | Method | GSM8K | MATH | AIME | Minerva | OlympiadBench | Avg. |
> | -- | -- | -- | -- | -- | -- | -- |
> | Backbone only   | 75.5 | 46.8 | 6.7 | 23.5 | 25.5 | 35.6 |
> | Ours          | 84.8 | 68.6 | 10.0 | 28.7 | 27.6 | 43.9 |
>
> **Llama translation**
> | Method | | EN → DE | | | | DE → EN | | |
> | -- | -- | -- | -- | -- | -- | -- | -- | -- |
> | | BLEU |CometKiwi |XComet |Avg. | BLEU |CometKiwi |XComet|Avg. |
> | Backbone only    | 25.68 | 79.87 | 92.34 | 65.96 | 32.46 | 82.26 | 95.28 | 70.00 |
> | Ours          | 27.94 | 81.71 | 93.95 | 67.87 | 37.88 | 82.92 | 96.02 | 72.27 |
>
> Thus, the remaining gap to external reports is attributable to our unified CoT prompt and evaluation protocol rather than additional errors.
>
> **Q2. Clarifying the architecture and implementation**
>
> Thank you for pointing this out. In the current version, Sec. 4.1 and Sec. 4.2 define UniR at the level of the log-probabilities and the GRPO objective, and Figure 1 depicts both the inference and training pipelines, but we agree that the step-by-step decoding procedure is implicit and could be clearer.
>
> At each decoding step, the backbone and the reasoning module process the same prefix, produce their own logits, and we add these logits and apply a softmax to obtain the next-token distribution. This logit-summed decoding is the only architectural modification. To remove any ambiguity, we introduce the following short pseudocode:
>
> ---
> **Algorithm 1: UniR logit-summed decoding**
>
> Input: Frozen Backbone Model $\pi_b$, Trained Reasoning Module policy $\pi_r$, Prompt $x$
>
> $y$ ← ∅
> for $t$ = 1...$T_\text{max}$ do
> $\quad$    $l_b$ ← logits($\pi_b$, $x$)
> $\quad$    $l_r$ ← logits($\pi_r$, $x$)
> $\quad$    $l_\theta$ ← $l_b$ + $l_r$
> $\quad$    $p_\theta$ ← softmax($l_\theta$)
> $\quad$    $y_t$ ← DecodeFrom($\pi_\theta$)  $\quad$   # greedy or sampling
> $\quad$    $x$ ← $x$ ⊕ $y_t$
> $\quad$    $y$ ← $y$ ⊕ $y_t$
> $\quad$    if $y_t$ = EOS then break
> return $y$

---

### Official Review · Reviewer_sxY5 · 2025-11-03

**Soundness:** 3
**Presentation:** 2
**Contribution:** 2
**Rating:** 4
**Confidence:** 5

**Summary:**

The paper introduces UniR, a lightweight reasoning module trained with verifiable rewards that can be plugged into frozen LLMs at inference by simply adding its logits to those of the backbone. The module is (i) architecture-agnostic (only tokenizer alignment required), (ii) composable (multiple UniRs can be summed for multi-task reasoning), and (iii) shown to transfer from small to much larger backbones without further tuning. Experiments on mathematical reasoning, machine translation, vision-language math, and medical risk prediction report gains over full-model GRPO fine-tuning while using far fewer trainable parameters and less memory.

**Strengths:**

Originality
- Proposes a clean, modular training objective mapping trajectory reward to token-level log-probabilities and uses additive logit fusion at inference which is a conceptually neat design.
- Inference-time composition via logit addition is derived from a KL-regularized multi-objective RL objective (Eq. 6), giving the method theoretical grounding absent in prior adapter ensembles.

Empirical Quality
- Consistent gains across 5 math benchmarks (up to +7.4 avg. over GRPO-full) and 2 translation directions (+1.8 BLEU) with ≤ 1.5 B trainable parameters.
- Strong weak-to-strong transfer: a 1.5 B module trained with a 3 B backbone improves a frozen 14 B Qwen-2.5 by +2.5 avg. points (Fig. 2).
- Composability demo: combining math + translation modules on German math→English task beats either specialist alone (Fig. 3).

Clarity & Reproducibility
- Complete hyper-parameter tables (App. A.1), reward definitions, and prompts (App. D) provided.
- Seeds and std-dev reported (10 runs, Tables 1–2).

Significance
- Enables cheap, plug-and-play specialization of proprietary or edge-deployed models without gradients through the backbone.

**Weaknesses:**

- Tokenizer-alignment prerequisite limits practicality. The paper claims “architecture-agnostic” but requires shared tokenizer (sec. 1, line 6). Is there a way to extend the method with tokenizer-mismatch adaptation strategy (e.g., embedding-level mapping layer trained with UniR) or can we quantify performance drop under vocabulary misalignment.

- The central identity (Eq. (4)), representing a trajectory reward as the sum of token log-probabilities of some policy, is assumed and only informally justified; Theorem 1 is informal and depends on strong assumptions (convergence of $\pi_\theta$ to $\pi_{\theta^*}$, uniqueness of optimum, and that π_r can exactly represent the reward) (Sec. 4.1 & 4.3). Can you provide a formal version of the theorem in the appendix with the details of the assumptions and proofs. The proof relies on the assumption that the guided policy has converged to the optimum. It would be beneficial to discuss the conditions under which this assumption holds.

- The paper compares to GRPO variants and LoRA but does not compare to other recent inference-time guidance / token-level reward approaches (beyond brief GenARM mention).

- Missing Related Work: The paper fails to cite or compare against a relevant line of work on plug-and-play modules for frozen transformers. Specifically, TART ("TART: A Plug-and-Play Transformer for Zero-Shot Task Adaptation", Bhatia et al., ICLR 2022) introduces a nearly identical architectural paradigm: a small, separately trained module whose outputs are combined with a frozen backbone model to achieve task adaptation. While TART's training objective and application (zero-shot classification) differ, the core concept of a "plug-and-play reasoner for frozen LLMs" is not entirely new.

**Questions:**

Please see the weakness section above for relevant questions. In addition,

- Can you show failure cases where composed modules hurt performance? What are the symptoms of negative interference?
- Code release plan -- will the code for this paper be released?
- Regarding composability, how do you envision setting the combination weights $\alpha_i$ in a real-world scenario without an expensive sweep for each new task combination? Have you considered methods for dynamically learning or setting these weights?

---

> ### Author Response · Authors · 2025-11-22
>
> **W1. Extension with tokenizer-mismatch adaptation strategy**
>
> We thank the reviewer for raising the tokenizer-alignment limitation. We fully agree that the shared-tokenizer requirement affects practicality. To directly examine tokenizer mismatch, we ran an additional tokenizer-alignment experiment based on EVA [1], summarized below and detailed in the appendix.
>
> EVA [1] introduces a cross-tokenizer alignment method that constructs a token-level similarity matrix between tokenizers and utilizes it to ensemble multiple LLMs effectively. To construct the similarity matrix, EVA first learns a linear mapping $W$ between the two models’ token embedding spaces using tokens that are shared between the two tokenizers. By projecting through this mapping, it computes similarities between all tokens from both models in a shared embedding space, obtaining a token-level similarity matrix between the two vocabularies. In our experiments, we adopt this procedure to construct the similarity matrix between Qwen and Llama families. Instead of using their ensemble strategy, we apply the UniR additive logit fusion rule on top of this matrix.
>
> Using this method, we compare two settings: (i) Qwen-3B + Llama-1B, and (ii) Qwen-3B + Llama-Unir-1B, where Llama-Unir-1B utilizes a trained reasoning module employing UniR method in the main results. (All results use greedy decoding.)
>
> | Method | GSM8K | MATH | AIME | Minerva | OlympiadBench | Avg. |
> | -- | -- | -- | -- | -- | -- | -- |
> | Qwen 3B 			| 75.5 | 46.8 | 6.7 | 23.5 | 25.5 | 35.6 |
> | Qwen 3B + Llama1B 	| 38.4 |14.0 | 3.3 | 4.8 | 3.0 | 12.7 |
> | Qwen 3B + Llama-Unir-1B   | 38.1 |28.4 | 6.7 | 10.3 | 8.1 | 18.3 |
>
> In the result, both EVA-base alignment performances are lower than using backbone alone; however, the setting with the trained UniR module (Avg.: 18.3) outperforms that with the untrained one (Avg.: 12.7). This indicates that, even under imperfect cross-tokenizer alignment, the UniR reasoning module can still guide the backbone LLM to better reasoning than its untrained counterpart. We added this result to the appendix, and we believe that more sophisticated tokenizer-alignment strategies are a promising direction for future work.
>
> [1] “Bridging the Gap between Different Vocabularies for LLM Ensemble”, NAACL, 2024
>
> **W2. Providing a Formal Theorem**
>
> We sincerely thank the reviewer for a critical concern regarding the theoretical foundation of our work. We would like to highlight that our intention was not to state Eq. (4) as a universal truth, but as a parameterization without loss of generality. We have revised our theoretical justification to provide a more rigorous foundation.
>
> Our revised argument proceeds in two steps:
> 1. We first prove that parameterizing the trajectory-level reward $r(x,y)$ as a log-probability $\beta \log \pi_r(y|x)$ (which decomposes into a sum of token-level log-probabilities by the chain rule) does not lose generality for the KL-regularized objective.
> 2. Using this justified parameterization, we re-frame Theorem 1 not as a result of assumed convergence, but as a formal interpretation of what the reasoning module $\pi_r(\cdot;\phi)$ learns when the guided policy $\pi_\theta$ is optimized towards the optimal policy $\pi_\theta^*$.
>
> We include it in the Appendix A in the revised version.

---

> ### Author Response · Authors · 2025-11-22
>
> **W3. Comparison of token-level reward or inference-time guidance**
>
> We thank the reviewer for constructive feedback. We agree that a comparison with other recent token-level reward and inference-time guidance approaches strengthens the paper.
>
> However, we would like to clarify a fundamental distinction between our approach and existing token-level reward methods. Most prior token-level approaches rely on an independently trained surrogate reward model to approximate token-wise signals [1, 2, 3]. In contrast, our method leverages a verifiable, model-free reward based on deterministic correctness checks, eliminating the need for a learned reward model. Given this structural difference, a direct comparison with surrogate-based methods is methodologically inappropriate.
>
> Instead, we have focused our comparative analysis on recent, concurrent inference-time guidance approaches (e.g., Reasoning Vectors [4] and RAST [5]) which represent the most relevant methodological alternatives to our model-free framework.
>
> 1.	**Reasoning Vectors**: Reasoning Vectors approach utilizes Task Arithmetic [6] to transfer reasoning ability within LLMs. To enhance a target model ($\theta_{target}$), the reasoning vector $v_{reason}$, as defined by the difference of the model parameter between a Reinforcement Learning model ($\theta_{GRPO}$) and a Supervised Fine-Tuning model ($\theta_{SFT}$), is added to target model’s parameters with certain scaling factor. This constitutes inference-time guidance by directly manipulating the model weights, fundamentally differing from logit-level approaches like UniR.
>
> 2.	**RAST**: RAST leverages the logit-level adjustments from smaller RL-tuned LLMs. RAST obtained a logit difference by subtracting the logits of a small base model from those of its RL-trained counterpart. This difference is then added to the target large model’s logit to guide the larger model. UniR and RAST are the most closely related methods as both utilize direct logit manipulation for reasoning transfer. However, UniR simplifies the dependency by training a single module ($\pi_r$) directly with verifiable rewards, avoiding the need for two small models to generate the signal.
>
> For comparison against UniR, we designed the setup to consider two controlled factors: (i) the space where the guidance signal is applied (parameter v.s. logit), and (ii) the number of models required to produce that signal. To isolate the effect of the guidance space, we compared UniR’s logit-space guidance with a parameter-space variant of Reasoning Vector, where we formed $v_{reason}$ from the GRPO-trained model and the Backbone model, and adding this vector to the Backbone with a scaling factor of 2. To isolate the effect of the number of guiding models within the logit-space, we compared RAST which relies on a three-model framework. All baselines were evaluated using Qwen 3B as the backbone and Qwen 1.5B as the small LLM with the same GRPO setting as in our paper. The results are summarized below:
>
> | Method | GSM8K | MATH | AIME | Minerva | OlympiadBench | Avg. |
> | -- | -- | -- | -- | -- | -- | -- |
> | Backbone-only	| 74.4 ± 1.0 | 44.2 ± 1.3 | 6.3 ± 3.7 | 19.2 ± 1.7 | 24.6 ± 0.9 | 33.7
> | Reasoning Vector [4]| 81.4 ± 0.7 | 55.7 ± 1.3  | 3.0 ± 2.5   | 24.9 ± 1.4 | 23.8  ± 0.5  | 37.8  |
> | RAST [5] | 76.1 ± 0.1 |  58.1 ± 0.9  | 3.3 ± 3.3  | 24.8 ± 0.8 | 19.2 ± 1.5 | 36.3 |
> | UniR 	| 84.5 ± 0.7 | 66.8 ± 1.4 |7.7 ± 2.7  | 26.3 ± 2.0 | 28.2 ± 0.8  | 42.7 |
>
> The performance comparison shows that the logit-level guidance provided by the single trained UniR module consistently yields superior performance compared to the results of parameter-level manipulation (Reasoning Vectors) or the contrastive logit-subtraction approach (RAST).
>
> [1] "Training a Helpful and Harmless Assistant with Reinforcement Learning from Human Feedback.", Arxiv, 2022.
>
> [2]  "TLCR: Token-level Continuous Reward for Fine-grained Reinforcement Learning from Human Feedback." ACL, 2024
>
> [3] "Dpo Meets PPO: Reinforced Token Optimization for RLHF.", ICML, 2025
>
> [4] "Reasoning Vectors: Transferring Chain-of-Thought Capabilities via Task Arithmetic.", Arxiv, 2025
>
> [5] “RAST: Reasoning Activation in LLMs via Small-model Transfer”, NeurIPS, 2025
>
> [6] “Editing Models with Task Arithmetic”, ICLR, 2023

---

> ### Author Response · Authors · 2025-11-22
>
> **W4. Missing Related Work**
>
> We thank the reviewer for pointing out TART, which is a highly relevant line of work. We agree that it shares the high-level architectural paradigm of a plug-and-play module combined with a frozen backbone. However, we would like to highlight fundamental differences that distinguish UniR's contribution:
>
> 1.	**Point of Interaction**: TART combines a reasoning module on top of the final layer representations from the base LLM and then processes these embeddings for classification. In contrast, UniR operates at the logit level. This allows UniR to guide or steer the base LLM's native generative process at each token step, rather than superseding it. This additive composability is theoretically grounded in a multi-objective RL framework (Eq. 6), which is not explored in TART.
> 2.	**Task Domain and Objective**: TART is designed for classification tasks, with its module trained on synthetic logistic regression data. UniR is designed for complex, generative reasoning tasks, such as mathematical problems, machine translation. Also, our module is not a classifier; it is trained via policy gradient to maximize trajectory-level verifiable rewards.
> We add these comparisons to our related work section to clarify UniR's unique contribution in the context of plug-and-play modules.
>
> **Q1. Failure Analysis**
>
> We acknowledge the reviewer’s concern regarding potential negative interference. In Figure 22 in Appendix F.5., we provide representative incorrect cases. Our analysis indicates that a semantic interpretation failure is more dominant than arithmetic error. To be specific, these failures reflect fundamental challenges in semantic-to-arithmetic translation, where the model struggles to correctly map linguistic constraints to mathematical formulations.
>
> **Q2. Code release plan**
>
> Yes. All code used in this submission is already available through the supplementary material. Upon publication, we will release a documented version on GitHub with easy-to-run scripts for reproducibility.
>
> **Q3. Dynamic Weight Adaptation**
>
> While we set a fixed $\alpha=1$ for our main experiments, we also explored a training-free dynamic weighting scheme precisely to address the challenge of avoiding task-specific hyperparameter tuning.
>
> We dynamically calculate weights based on the confidence of the backbone and reasoning models. We define a model's confidence as the maximum probability value in its output distribution. $c_{B} = max(P_B), c_{r} = max(P_r)$, where $P_B$ and $P_r$ are the probability distributions derived from the last token logits of the backbone model and the reasoning module, respectively. We then calculate final logits as a weighted sum using these normalized confidence scores:
> $$w_B=\frac{c_{B}}{c_{B}+c_{r}}$$
> $$w_r=\frac{c_{r}}{c_{B}+c_{r}}$$
> $$logits'=w_B \cdot logits_B + w_r \cdot logits_r$$
> Intuitively, this formulation works as follows: when the reasoning module is uncertain (i.e., $c_r$ is low), its weight $w_r$ decreases, and the final output relies more heavily on the backbone model's logits and vice versa. This mechanism provides an adaptive composition without requiring additional optimization.
>
> We evaluated this dynamic weighting using Qwen-3B-Instruct as the backbone and Qwen-1.5B-Instruct as the reasoning module. The results are summarized below:
>
> | Method | GSM8K | MATH | AIME | Minerva | OlympiadBench | Avg. |
> | -- | -- | -- | -- | -- | -- | -- |
> | GRPO Full | 81.3 ± 0.5 |62.9 ± 1.3  | 4.7 ± 2.3 |25.0 ± 1.0  | 27.5 ± 1.3 | 40.3 |
> | UniR 	| 84.5 ± 0.7 | 66.8 ± 1.4 |7.7 ± 2.7  | 26.3 ± 2.0 | 28.2 ± 0.8  | 42.7 |
> | UniR (dynamic w)   | 83.5 ± 0.7 | 65.0 ± 1.8 | 5.0 ± 2.8  | 25.48 ± 0.7  | 27.0 ± 0.7 | 41.19  |
>
> Dynamic weighting achieves better performance than the GRPO-based method and remains comparable to the original UniR. These results suggest that confidence-based weighting offers an efficient and practical method to modulate compositional contributions, as it operates in a training-free manner and eliminates the need for task-specific hyperparameter tuning. However, the current definition of confidence for dynamic weight adaptation may have a room for improvement. As a direction for future work, we plan to investigate fully adaptive composition mechanisms.

---

### Author Response · Authors · 2025-11-22
**General Response**

We sincerely thank the reviewers for their constructive and detailed feedback. We are encouraged by the consensus on our novelty and modular design (sxY5, xWTC, 8VRM), effectiveness across model scales (sxY5, PpAD, 8VRM), and clarity of presentation (PpAD, 8VRM).

In response to valuable suggestions regarding comparative baselines, theoretical rigor, and practical constraints, we have performed extensive new experiments and analyses. Below is a summary of the major revisions included in the updated manuscript.

**1. Strengthened Comparative Analysis.** To address the request for broader comparisons (Reviewer sxY5), we evaluated UniR against recent inference-time guidance methods, including Reasoning Vectors and RAST.

* **Result:** UniR's logit-space guidance consistently outperforms parameter-space manipulation (Reasoning Vectors) and contrastive logit subtraction (RAST), confirming the superiority of our direct verifiable reward training.
* **Baselines:** We also corrected the baseline numbers for Llama-3.2-3B and provided a detailed breakdown of how our zero-shot CoT prompting and decoding strategies influence performance compared to public leaderboards (Reviewer xWTC). This has been corrected and reflected in Table 1.

**2. Expanded Practicality and Generalization.** We conducted additional experiments to address practical deployment concerns raised by Reviewers sxY5 and PpAD.
* **Tokenizer Mismatch:** We demonstrated that UniR remains effective even with mismatched tokenizers (Qwen backbone + Llama module) by utilizing cross-tokenizer alignment (EVA), showing significant gains over the untrained baseline.
* **Dynamic Weighting:** We introduced a training-free dynamic weighting scheme based on model confidence, eliminating the need for manual $\alpha$ tuning while maintaining performance comparable to the optimal fixed weight.
* **Computational Efficiency:** We added a detailed cost analysis, highlighting that UniR significantly reduces VRAM usage and offers higher training throughput compared to GRPO-Full.

**3. Theoretical Rigor and Architectural Clarity.** We have refined the theoretical and algorithmic descriptions to remove ambiguity (Reviewers sxY5, xWTC).
* **Formalization:** We revised the justification for our trajectory-level reward parameterization and re-framed Theorem 1 to provide a more rigorous theoretical foundation for the reasoning module's objective. This information is located in Appendix A.
* **Algorithm:** We included formal pseudocode (Algorithm 1) to explicitly define the logit-summed decoding procedure, ensuring reproducibility.

**4. Deepened Mechanistic Analysis.** We investigated the limits and scalability of our modular design (Reviewers PpAD, 8VRM).
* **Minimal Module:** We experimented with training only the LM head as a reasoning module. While it improves performance, results confirm that the multi-layer architecture of a small LLM is beneficial for capturing complex reasoning guidance.
* **Composition Scaling:** We explored composing up to three reasoning modules (medical tasks). The results provide improved insights into the trade-offs between synergistic improvements and task interference.

We believe these revisions significantly strengthen the paper and provide robust evidence that UniR is a scalable, effective, and theoretically grounded solution for LLM reasoning. We thank the reviewers again for helping us improve the quality of this work.

We have provided detailed responses to each reviewer below and look forward to any further discussion.

---

> ### Comment · Reviewer_xWTC · 2025-11-25
>
> >Below is a summary of the major revisions included in the updated manuscript.
>
> What updated manuscript? I do not see any updates to the manuscript.

---

> ### Author Response · Authors · 2025-11-26
>
> We have already uploaded the revised manuscript with updates highlighted in blue color. Please check the revised version. Specifically, the following sections and components have been updated:
>
> - 1 Introduction - (PpAD) Clarifying our contribution statement.
>
> - 2 Related Work - (sxY5) Adding a comparison with the TART paper.
>
> - Table 1 - (xWTC) Correcting backbone baseline numbers for Llama-3.2-3B.
>
> - 5.3 and Figure 3 - (PpAD) Updating the comparison with baseline performance.
>
> - Appendix A - (sxY5) Providing a formal Theorem.
>
> - Appendix Table 7 - (8VRM) Clarifying the value of the coefficient $\alpha$.
>
> - Appendix F.5 and Figure 22 - (sxY5) Adding examples of failure cases.
>
> The details related to each update have also been documented in the review comments.

---

> > ### Comment · Reviewer_xWTC · 2025-11-26
> >
> > But the revisions tab says there are no revisions? It says "No revisions to display."
> >
> > Maybe this is an OpenReview bug.

---

> > > ### Author Response · Authors · 2025-11-26
> > >
> > > It might indeed be an OpenReview display issue.
> > >
> > > You should be able to view the updated manuscript immediately by clicking on the PDF icon located right at the top of this page.

---

### Meta-Review · Area_Chair_fRbq · 2025-12-26

**Summary:**

This paper introduces UniR, a lightweight, modular reasoning module designed to be added to frozen large language models (LLMs) via logit-space composition at inference time. The central contribution is a general and architecture-agnostic plug-and-play framework that enables efficient and composable reasoning capabilities without modifying the base model. The authors train UniR using a verifiable reward objective and demonstrate its ability to transfer from smaller to larger LLMs, as well as its composability across tasks.

Reviewers highlighted the novelty of the approach, particularly the logit addition mechanism, weak-to-strong generalization, and multi-task composition. The empirical results show substantial gains over fine-tuning baselines (GRPO-Full and GRPO-LoRA) in diverse domains, including math reasoning, translation, and clinical prediction tasks.

Most reviewers initially rated the paper as borderline due to missing details and some key issues, e.g., theoretical gaps (Theorem 1 formalization), practical concerns (tokenizer mismatch, cost), and empirical omissions (comparisons with RAST, Reasoning Vectors, TART). The rebuttal attempted to address the concerns, but there may still exist limitations in theoretical rigor and practical generalizability. Further, the baseline numbers seem a bit skeptical to me (reviewer xWTC also raised the concern). For example, I checked out one of my own previous Qwen2.5-3B models trained with full GRPO, which achieves 65.8% accuracy on MATH-500, while the authors report a 62.9% for the full GRPO baseline and 66.8% for their model. The settings might not be directly comparable between mine and theirs. But I'm still a bit skeptical about whether the authors correctly set up the baseline GRPO training.

From the discussion, I'd recommend a rejection of the paper (admitting its method novelty). I recommend that the authors to: 1) do a further investigation of all of the experimental setups/baselines; 2) address cross-tokenizer alignment issues. 3) add more analysis of the dynamic weighting scheme.

**Reviewer Concerns:**

- Although cross-tokenizer alignment via EVA was explored, performance remains suboptimal. Future work should develop more robust alignment or embedding mapping strategies for full architecture agnosticism.

- The dynamic weighting scheme based on model confidence shows promise, but could benefit from further analysis or learning-based adaptation for multi-module inference.

- While the authors added a formal Theorem 1, assumptions about reward representability and convergence could be made more precise or validated empirically.

- Inconsistent baseline numbers (e.g., for Llama-3.2-3B on MATH).

**Reviewer Scores:**

sxY5 (4) → Would likely raise to 6 or maintain 4 based on the formalized theory, tokenizer experiments, and stronger baselines.

xWTC (6) → Would likely maintain 6, now reassured about reproducibility and baseline correctness.

PpAD (6) → Would maintain 6; main concerns about efficiency and module design were reasonably addressed.

8VRM (6) → Would maintain 6, though likely still curious about the scalability of dynamic composition.

---

### Decision · Program_Chairs · 2026-01-26

Reject